



# An Analysis of New Particle Formation (NPF) at Thirteen European Sites

**Dimitrios Bousiotis[1], Francis D. Pope[1], David C. Beddows[1],
Manuel Dall'Osto[2], Andreas Massling[3], Jacob Klenø Nøjgaard[3],
Claus Nørdstrom[3], Jarkko V. Niemi[4], Harri Portin[4], Tuukka Petäjä[5],
Noemi Perez[6], Andrés Alastuey[6], Xavier Querol[6], Giorgos Kouvarakis[7],
Stergios Vratolis[8], Konstantinos Eleftheriadis[8], Alfred Wiedensohler[9], Kay Weinhold[9],
Maik Merkel[9], Thomas Tuch[9] and Roy M. Harrison[1]\*†**

**[1]Division of Environmental Health and Risk Management
School of Geography, Earth and Environmental Sciences
University of Birmingham, Edgbaston, Birmingham B15 2TT, United Kingdom**

**[2]Institute of Marine Sciences
Passeig Marítim de la Barceloneta, 37-49, E-08003, Barcelona, Spain**

**[3]Department for Environmental Science, Aarhus University, DK-400, Roskilde, Denmark**

**[4]Helsinki Region Environmental Services Authority (HSY)
FI-00066 HSY, Helsinki, Finland**

**[5]Institute for Atmospheric and Earth System Research (INAR) / Physics, Faculty of Science University
of Helsinki, Finland**

**[6]Institute of Environmental Assessment and Water Research (IDAEA - CSIC)
08034, Barcelona, Spain**

**[7]Environmental Chemical Processes Laboratory (ECPL), Department of Chemistry
University of Crete, 70013, Heraklion, Greece**

**[8]Environmental Radioactivity Laboratory
Institute of Nuclear and Radiological Science & Technology, Energy & Safety
NCSR Demokritos, Athens, Greece**

**[9]Leibniz Institute for Tropospheric Research (TROPOS),
Permoserstr. 15, 04318 Leipzig, Germany**

---

\* To whom correspondence should be addressed (Email: r.m.harrison@bham.ac.uk)
†Also at: Department of Environmental Sciences / Center of Excellence in Environmental Studies, King Abdulaziz
University, PO Box 80203, Jeddah, 21589, Saudi Arabia



**ABSTRACT**
New particle formation (NPF) events occur almost everywhere in the world and can play an
important role as a particle source.  The frequency and characteristics of NPF events vary spatially
and this variability is yet to be fully understood. In the present study, long term particle size
distribution datasets (minimum of three years) from thirteen sites of various land uses and climates
from across Europe were studied and NPF events, deriving from secondary formation and not
traffic related nucleation, were extracted and analysed. The frequency of NPF events was
consistently found to be higher at rural background sites, while the growth and formation rates of
newly formed particles were higher at roadsides, underlining the importance of the abundance of
condensable compounds of anthropogenic origin found there. The growth rate was higher in
summer at all rural background sites studied. The urban background sites presented the highest
uncertainty due to greater variability compared to the other two types of site. The origin of
incoming air masses and the specific conditions associated with them greatly affect the
characteristics of NPF events. In general, cleaner air masses present higher probability for NPF
events, while the more polluted ones show higher growth rates. However, different patterns of NPF
events were found even at sites in close proximity (< 200 km) due to the different local conditions
at each site. Region-wide events were also studied and were found to be associated with the same
conditions as local events, although some variability was found which was associated with the
different seasonality of the events at two neighbouring sites. NPF events were responsible for an
increase in the number concentration of ultrafine particles of more than 400% at rural background



sites on the day of their occurrence. The degree of enhancement was less at urban sites due to the
increased contribution of other sources within the urban environment. It is evident that, while some
variables (such as solar radiation intensity, relative humidity or the concentrations of specific
pollutants) appear to have a similar influence on NPF events across all sites, it is impossible to
predict the characteristics of NPF events at a site using just these variables, due to the crucial role of
local conditions.
**Keywords:** Nucleation; New Particle Formation; Ultrafine Particles; Roadside; Urban Background;
Rural



## 1.     INTRODUCTION


Ultrafine particles (particles with diameter smaller than 100 nm), while not yet regulated, are
believed to have adverse effects upon air quality and public health (Atkinson et al., 2010; Politis et
al., 2008; Tobías et al., 2018), as well as having a direct or indirect effect on atmospheric properties
(Makkonen et al., 2012; Seinfeld and Pandis, 2012). The source of ultrafine particles can either be
from primary emissions (Harrison et al., 2000; Masiol et al., 2017), including delayed primary
emissions (Hietikko et al., 2018; Olin et al., 2020; Rönkkö et al., 2017), or from secondary
formation from gaseous precursors (Brean et al., 2019; Chu et al., 2019; Kerminen et al., 2018;
Kulmala et al., 2004a; Yao et al., 2018), which is considered as an important source of CCN in the
atmosphere (Dameto de España et al., 2017; Kalivitis et al., 2015; Spracklen et al., 2008). For the
latter, while the process of formation of initial clusters that subsequently lead to particle formation
has been extensively studied (Dal Maso et al., 2002; Kulmala et al., 2014; Riipinen et al., 2007;
Weber et al., 1998), there is no consistent explanation of the factors which determine the occurrence
and development of NPF events in the atmosphere. Additionally, events that resemble NPF, with
the initial particles deriving from primary emissions, especially close to traffic sources (Rönkkö et
al., 2017), have been also reported but these are out of the scope of the present study.

A large number of studies both in laboratories and in real world conditions have been conducted to
either describe or explain the mechanisms that drive NPF events. The role of meteorological
conditions, such as solar radiation intensity (Kumar et al., 2014; Shi et al., 2001; Stanier et al.,



2004) and relative humidity (Li et al., 2019; Park et al., 2015), are well documented, while great
diversity was found for the effect of other meteorological factors such as the wind speed (Charron et
al., 2008; Németh and Salma, 2014; Rimnácová et al., 2011) or temperature (Jeong et al., 2010;
Napari et al., 2002). There are also influences of atmospheric composition, with the positive role of
low condensation sink and concentrations of pollutants such as $NO_x$ upon NPF event occurrence
being widely agreed upon (Alam et al., 2003; Cheung et al., 2013; Kerminen et al., 2004; Wang et
al., 2014; Wehner et al., 2007). Contrary to that, while the indirect role of $SO_2$ is well established in
the nucleation process, via the formation of new clusters of $H_2SO_4$ molecules (Boy et al., 2005; Iida
et al., 2008; Kulmala et al., 2005; Sipila et al., 2010; Xiao et al., 2015), uncertainty exists in the role
that different concentrations of $SO_2$ play in the occurrence of NPF events in real world atmospheric
conditions (Alam et al., 2003; Dall'Osto et al., 2018; Wonaschütz et al., 2015; Woo et al., 2001).
Ammonia is known to enhance the formation of initial clusters (Korhonen et al., 1999; Ortega et al.,
2008; Schobesberger et al., 2015), and volatile organic compounds are regarded as the main drivers
of the growth of the newly formed particles (Kulmala et al., 2013; Riccobono et al., 2014; Tröstl et
al., 2016). NPF events in different locations do not appear to follow consistent trends with the
concentrations of these compounds and meteorological parameters (McFiggans et al., 2019;
Minguillón et al., 2015; Riipinen et al., 2007), though links between NPF events and sulphuric acid
vapour concentrations (Petäjä et al., 2009; Weber et al., 1995) and organics (Bianchi et al., 2019;
Ehn et al., 2014) have been reported.





It is evident that NPF events and their development are complex, and local conditions play an
important role in their variability. Many studies have attempted to explain this variability by
analyzing multiple datasets from wider areas. Studies in the UK (Bousiotis et al., 2019; Hama et al.,
2017), Spain (Brines et al., 2014; Carnerero et al., 2018; Dall'Osto et al., 2013; Minguillón et al.,
2015), Hungary (Németh and Salma, 2014; Salma et al., 2014, 2016), Greece (Kalkavouras et al.,
2017; Siakavaras et al., 2016), Germany (Costabile et al., 2009; Ma and Birmili, 2015; Sun et al.,
2019) and China (Peng et al., 2017; Shen et al., 2018; Wang et al., 2017) have attempted to explain
the differences found in NPF event conditions and variability between different sites in close
proximity, while larger scale studies using descriptive (Brines et al., 2015; Hofman et al., 2016;
Jaatinen et al., 2009; Kulmala et al., 2005) or statistical methods (Dall'Osto et al., 2018; Rivas et
al., 2020) have provided insights into the effect of the variability of parameters that are considered
to play an important role in the occurrence and development of NPF events on a broader scale.

The present study, combining thirteen long term datasets (minimum of three years) from different
countries across Europe and combined with the results from a previous study in the UK, attempts to
elucidate the effect of the local conditions on NPF event characteristics (frequency of NPF events,
formation rate and growth rate) both for sites in close proximity (< 200 km), and by
intercomparison of sites on a continental scale in order to find general trends of the variables that
affect the characteristics and development of NPF events on a larger scale. Finally, the effect of





NPF events upon the ultrafine particle number concentrations was calculated, providing insight to
the potential of NPF events to influence the local air quality conditions in all areas studied.

**2.    DATA AND METHODS**
**2.1    Site Description and Data Availability**
In the present study, particle number size distribution data from 13 sites in Europe (Figure 1) are
analysed in the size range 3 nm $< D_p <$ 1000 nm. A detailed list of the site locations and the data
available for each is found in Table 1. Average meteorological conditions and concentrations of
chemical compounds for all sites are found in Tables S1 and S2 respectively; their seasonal
variation is found in Table S3.

**2.2    Methods**
**2.2.1   NPF event selection**
The identification of NPF events was conducted manually using the criteria set by Dal Maso et al.
(2005). According to these, a NPF event is considered to occur when:

•    a distinctly new mode of particles appears in the nucleation range,
•    this new mode prevails for some hours,
•    the new mode shows signs of growth.



The NPF events extracted using this method are then classified into classes I or II depending on the
level of confidence. Class I (high confidence) is further classified as Ia and Ib, with class Ia
containing the events that both present a clear formation of a new mode as well as a distinct growth
of this mode, while Ib includes those with a less distinct formation and development. In the present
study, only the events classified as Ia were used as they are considered as more suitable for study.
As the growth criterion is not fully defined, in the present study a minimum growth rate of 1 nm h$^{-1}$
is required for NPF events to be considered. The events found using this method should not be
confused with the formation and growth of particles deriving from primary emissions next to
pollution sources, such as traffic. While to an extent the particle formation found can be biased by
primary emissions (especially at roadside sites), great effort was made using additional data, such as
atmospheric composition data, to not include any incidents of traffic related nucleation.

**159   2.2.2   Calculation of condensation sink, growth rate, formation rate, Nucleation Strength**

**160         Factor (NSF) and NPF event probability**

The calculation of the condensation sink was made using the method proposed by Kulmala et al.
(2001). The condensation sink (CS) is calculated as:

$$CS = 4\pi D_{vap} \sum \beta_M \, r \, N$$





where r and N are the radius and the number concentration of the particles and $D_{vap}$ is the diffusion
coefficient, calculated for $T = 293$ K and $P = 1013.25$ mbar, according to Poling et al. (2001):

$$D_{vap} = 0.00143 \cdot T^{1.75} \frac{\sqrt{M_{air}^{-1} + M_{vap}^{-1}}}{P \left( D_{x,air}^{\frac{1}{3}} + D_{x,vap}^{\frac{1}{3}} \right)^2}$$

where M and $D_x$ are the molar mass and diffusion volume for air and $H_2SO_4$. $\beta_M$ is the Fuchs
correction factor calculated from Fuchs and Sutugin (1971):

$$\beta_M = \frac{1 + K_n}{1 + \left( \frac{4}{3a} + 0.377 \right) K_n + \frac{4}{3a} K_n^2}$$

$K_n$ is the Knudsen number, defined as $Kn = 2\lambda_m/d_p$, with $\lambda_m$ being the mean free path of the gas.

The growth rate of the newly formed particles is calculated according to Kulmala et al. (2012), as

$$GR = \frac{D_{P_2} - D_{P_1}}{t_2 - t_1}$$





for the size range between the minimum available particle diameter up to 30 nm. For the calculation
of the growth rate, the time considered was from the start of the event until a) growth stopped, b)
GMD reached the upper limit set or c) the day ended.

The formation rate J was calculated using the method proposed by Kulmala et al. (2012) in which:

$J_{d_p} = \dfrac{dN_{d_p}}{dt} + CoagS_{d_p} \times N_{d_p} + \dfrac{GR}{\Delta d_p} \times N_{d_p} + S_{losses}$

where $CoagS_{dp}$ is the coagulation rate of particles of diameter $d_p$, calculated by:

$CoagS_{d_p} = \displaystyle\int K(d_p, d'_p)\, n(d'_p) dd'_p \cong \sum_{d'_p = d_p}^{d'_p = max} K(d_p, d'_p)\, N_{d_p}$

as proposed by Kerminen et al. (2001). $K(d_p, d'_p)$ is the coagulation coefficient of particle sizes $d_p$
and $d'_p$. $S_{losses}$ accounts for the additional loss terms (i.e. chamber walls), not considered here. Initial
particle formation starts at about $1.5 \pm 0.4$ nm (Kulmala et al., 2012). The formation rate calculated
here refers to particles in the atmosphere that reached the diameter of 10 nm during NPF events for
uniformity reasons. This means that these particles were formed earlier during the day of the events,
survived and grew to this size later in the day. Furthermore, due to the effect of the morning rush





which biased the results at roadsides, the averages are calculated for the time window between 9:00
to 15:00 ($\pm$ 3 hours from noon, when $J_{10}$ peaked in the majority of the events). This was done for all
the sites in this study for consistency.

The Nucleation Strength Factor (NSF) proposed by Nemeth and Salma (2014) is a measure of the
effect of NPF events on ultrafine particle concentration. It can either refer to the effect of NPF
events on the day of their occurrence, calculated by:

$$NSF_{NUC} = \frac{\left(\frac{N_{\text{smallest size available}-100nm}}{N_{100nm\ -\text{largest size available}}}\right)_{\text{nucleation days}}}{\left(\frac{N_{\text{smallest size available}-100nm}}{N_{100nm-\text{largest size available}}}\right)_{\text{non-nucleation days}}}$$

or their overall contribution on the ultrafine particle concentrations at a site calculated by:

$$NSF_{GEN} = \frac{\left(\frac{N_{\text{smallest size available}-100nm}}{N_{100nm-\text{largest size available}}}\right)_{\text{all days}}}{\left(\frac{N_{\text{smallest size available}-100nm}}{N_{100nm-\text{largest size available}}}\right)_{\text{non-nucleation days}}}$$

The NPF event probability is a simple metric of the probability of NPF events calculated by the
number of NPF event days divided by the number of days with available data for the given group





(temporal, wind direction etc.). Finally, it should be mentioned that all the results presented are
normalised according the seasonal data availability for each site, based upon the expression:

$NPF_{probability} = \dfrac{N_{NPF\ event\ days\ for\ group\ of\ days\ X}}{N_{days\ with\ available\ data\ for\ group\ of\ days\ X}}$

**3.    RESULTS AND DISCUSSION**
**3.1    Denmark**
NPF events occurred at all three sites with available data with a similar frequency for the urban sites
(5.4% for DENRO and 5.8% for DENUB) and higher for the rural DENRU site (7.9%), for the nine
year period of this study (2008 – 2017). For the DENRO and DENRU sites the seasonal variation
favoured summer, while at DENUB a higher frequency of events was found for spring (Figure 2).
The growth rate was found to be higher at the DENRO site at 4.45±1.87 nm h$^{-1}$ and it was similar
for the other two sites (3.19±1.43 for DENRU and 3.19±1.45 for DENUB) nm h$^{-1}$ (Figure 3),
though the peak was found in different seasons (Figure 5), coinciding with that of the frequency of
NPF events (the highest average for DENRO was found for winter but it was only for a single event
that occurred in that season). As for the within-week variation of the events, there is an increasing
probability of NPF events to occur on weekends than weekdays going from the rural background
site to the roadside site (Figure 4). Interesting (and probably coincidental) is the increased
frequency of NPF events found at all sites on Thursday among the weekdays. J$_{10}$ was found to be
broadly similar at the rural and urban background sites and higher at DENRO (Figure 6), favoured





by different seasons at each site (summer at DENRU, spring at DENUB though with minimum
differences and autumn at DENRO) (Figure 7).

In general, pollutant concentrations were found to be lower on event days for all sites (apart from
$O_3$), including the secondary pollutants and minerals (apart from marine related elements like Na,
Cl and Mg – data not included) where data was available (Table S2). Among the compounds with
lower concentrations on NPF event days was $SO_2$ (for the sites with available data), possibly due to
being in sufficient concentrations for not being a limiting factor in the occurrence of NPF events,
while higher concentrations are associated with increased pollution conditions which may suppress
the occurrence of the events.

The meteorological conditions that prevailed on NPF event days (Table S1) were higher incoming
solar radiation, wind speed and temperature and lower relative humidity compared to average
conditions (consistently at all sites and significant for all ($p < 0.001$) except wind speed). As
meteorological conditions were available from the urban background site (the variation between the
rural and urban sites should not be great since they are about 25 km away from each other), the
average conditions for the three sites are almost the same with the only variability being the data
availability among the sites. Thus, the more common wind directions in the area are southwesterly;
for all sites though the majority of NPF events are associated with direct westerly and northwesterly
winds, similar to the findings of Wang et al. (2013) for the same site, which are those with the





lowest concentrations of pollutants and condensation sink for all sites, probably being of marine
origin as elemental concentrations showed an increased presence of Na, Cl and Mg (results not
included). The wind directions with the highest probability for NPF events present low growth rates
and vice versa (Table S4), though it was proposed by Kristensson et al. (2008) that there is a
possibility for events observed at the nearby Vavihill site in Sweden with northwesterly winds to be
associated to the emissions of specific ship lanes that pass from that area. Wind direction sectors
with higher concentrations of OC coincide with higher growth rates at DENRO, while this
variability is not found at DENRU possibly showing that different compounds and mechanisms take
part in the growth process of the newly formed particles (Kulmala et al., 2004b).

As mentioned earlier, DENUB although close to the DENRO site has different seasonal variation of
NPF events with a marginally lower frequency in summer compared to the other two Danish sites,
which have almost the same seasonal variation of NPF events. At DENUB, a strong presence of
particles in the size range of about 50 – 60 nm is observed (Figure S1), especially during summer
months, increasing the condensation sink in the area (this enhanced mode of particles is visible at
DENRO as well, but its effect is dampened due to the elevated particle number concentrations in
the other modes). This mode is probably part of the urban particle background. The strongest source
though at DENUB appears to be from the east and consistently appears at both urban sites; this
sector is where both elevated pollutant concentrations and condensation sink are found. In this
sector, there are two possible local sources, either the port located 2 km to the east or the power





plants located at a similar distance (or both). In general, both stations are located only a few
kilometres away from the Øresund strait, a major shipping route. Studying the SMPS plots it can be
seen that NPF events at DENUB especially in summer tend to start but are either suppressed after
the start or have a lifetime of a couple of hours before the new particles are scavenged or evaporate.
While this might explain to an extent the frequency and variability of NPF events at this site, the
balance between the condensation sink and the concentration of condensable compounds is
highlighted. While at DENRO the condensation sink is considerably higher than at DENUB and the
effect of the aforementioned mode of particles is present on both, the occurrence and development
of NPF events at DENRO are more pronounced in the data due to the higher concentrations of
condensable compounds.

**3.2     Germany**
A higher frequency of NPF events was found for each type of site in Germany compared to the
other countries in this study, for the three year period of this study (2008 – 2011). The background
sites had NPF events for more than 17% of the days, while the roadside had a lower frequency of
about 9%, with a seasonal variability favouring summer at all sites (Figure 2). It should be noted
though that due to the lack of spring and summer data for the first two years at GERRO, the
frequency of events is probably a lot higher and the seasonal variation should further favour these
seasons. Similarly, all sites had higher growth rates compared to sites of the same type in other
areas of this study, with GERRU having $4.34\pm1.73$ nm h$^{-1}$, GERUB $4.24\pm1.69$ nm h$^{-1}$ and GERRO





$5.17\pm2.20$ nm h$^{-1}$ (Figure 3). While the difference between GERRU and GERUB is not statistically
significant, there is a significant difference for GERRO ($p < 0.005$).  Higher growth rates were
found in summer compared to spring for all sites (Figure 5). Specifically for the roadside though,
the highest average growth rates were found in autumn, which may be either a site-specific feature
or an artefact of the limited number of events in that season (total of 11 NPF events in autumn). No
substantial within-week variation was found for any of the sites in this country (Figure 4), a feature
that is expected mainly at background sites. For GERRO, this may be due to not being as polluted
as other sites of the same type, having an average condensation sink comparable to that of urban
background sites. $J_{10}$ at the German sites was also the highest among the sites of this study (Figure
6), increasing from the GERRU to GERRO. It was found to be higher in summer for the
background sites and in autumn for GERRO (Figure 7).

Compared to the average conditions, a higher temperature and solar radiation were found on NPF
event days, while wind speed and relative humidity were lower at all sites (Table S1). The wind
profile is different between the urban and the rural sites, with mainly northeasterly and
southwesterly winds at the rural site and a more balanced profile for the urban sites. This difference
is probably due to differences in the local topography. For the urban sites the majority of NPF
events are associated with easterly winds (to a lesser extent westerly as well for GERRO). At
GERUB, along with the increased frequency of NPF events the highest average growth rate is also
found with easterly wind directions (though the differences are rather small). At GERRO the



frequency and growth rate appear to be affected by the topography of the site. Eisenbahnstraße is a
road with an axis at almost 90° – 270° and although the H/W ratio (surrounding buildings' height to
width ratio) is not high, the effect of a street canyon vortex is observed (Voigtländer et al., 2006).
Possibly as a consequence of this, the probability of NPF events is low for direct northerly and
southerly winds, although there are high growth rates of the newly formed particles (highest growth
rates observed with southerly winds, associated with cleaner air).

At GERRU an increased probability of NPF events and growth rate are also found for wind
directions from the easterly sector, although these are not very frequent for this site (Table S4). For
this site chemical composition data for $PM_{2.5}$ and $PM_{10}$ are available, and it is found that the
generally low (on average) concentrations of pollutants (such as elemental carbon, nitrate and
sulphate) in general are elevated for wind directions from that sector. This is also reported for the
Melpitz site (GERRU) by Jaatinen et al. (2009) and probably indicates that in a relatively clean
area, the presence of low concentrations of pollutants may be favourable in the occurrence and
development of NPF events, as in general pollutant concentrations are lower on NPF event days
compared to average conditions. Another interesting point is the concentration of organic carbon at
the site (average of 2.18 µg m$^{-3}$ in $PM_{2.5}$), having the highest average concentration among the rural
background sites studied. As other pollutant concentrations are relatively low at this site, it is
possible that a portion of this organic carbon is of biogenic origin, considering also that the area is
largely surrounded by forests and green areas, with a minimal effect of marine air masses (as





indicated by the low marine component concentrations – data not included) and possibly pointing to
increased presence of BVOCs. The increased presence of organic species at GERRU may explain to
some extent the increased frequency of NPF events as well as the highest growth and formation
rates found among the sites of this study.

**3.3    Finland**
NPF events at the sites studied in Finland presented the most diverse seasonal variation, peaking at
the background sites in spring and at the roadside in summer (Figure 2). The frequency of NPF
events at FINRU was higher (8.66%) for the years with available data (2008 – 2011 & 2015 –
2018), while being less at the urban sites (4.97% at FINUB and 5.20% at FINRO) for the three
years with available data for each (2008 – 2011 & 2015 - 2018 for FINUB and 2015 – 2018 for
FINRO). Growth rates were similar at the background sites ($2.91\pm1.68$ nm h$^{-1}$ at FINRU and
$2.87\pm1.33$ nm h$^{-1}$ at FINUB), peaking in summer months, similar to the findings of (Yli-Juuti et al.,
2011), while the peak for FINRO (growth rate at $3.74\pm1.48$ nm h$^{-1}$) was found in spring, though the
differences between the seasons for this site were rather small (Figures 3 and 5). Strong within-
week variation favouring weekends is found for the roadside, while no clear variation was found for
the other two sites (Figure 4). This may be due to either the higher condensation sink during
weekdays that suppresses the events or the dominant impact of the traffic emissions which could
make the detection of NPF events harder. $J_{10}$ was the highest at FINRO, peaking in autumn for both





urban sites (with small differences with spring), while FINRU presented the highest $J_{10}$ in summer
(Figures 6 and 7).

For all sites of this study in Finland, NPF events were consistently associated with lower relative
humidity and higher solar radiation (Table S1). At the background sites temperature was found to
be lower on NPF event days compared to the average conditions, whereas it was found higher for
FINRO associated with the different seasonality of the events. No significant differences were
found for the wind speed on NPF events for all sites. There are though some significant differences
in the wind conditions for NPF events compared to average conditions. At FINRU, NPF events
were more common with northerly wind directions, as was also found by Nieminen et al. (2014)
and Nilsson et al. (2001). This is probably due to the lower condensation sink which can be
associated with the lower relative humidity also found for incoming winds from that sector and also
explains the lower temperatures found with NPF events at this site (Table 4). Similarly, at FINUB
NPF events were favoured by wind directions from the northerly sector, while there is almost a
complete lack of NPF on southerly winds. This is due to its position at the north of both the city
centre and the harbour, though winds from that sector are not common in general for that site.
Finally, the wind profile for NPF events at FINRO also favours northerly winds with an almost
complete absence of southerly winds probably due to the elevated pollutant concentrations and
condensation sink associated with them.





At all sites, NPF event days had a lower condensation sink compared to the average for the site, as
well as lower concentrations of pollutants (apart from $O_3$) where data was available (Table S2). The
seasonal variation of NPF events in Finland favouring spring, was explained by earlier work as the
result of the seasonal variation of $H_2SO_4$ concentrations (Nieminen et al., 2014), which in the area
peak in spring. The variation of $H_2SO_4$ concentrations is directly associated with $SO_2$ concentrations
in the area, which follow a similar trend. The seasonal variation of NPF events at FINRO though
cannot be explained by the variation of $H_2SO_4$ in the area. $SO_2$ concentrations, which were available
only for the nearby urban background site at Kalio (about 3 km away from FINRO) and may
provide information upon the trends of $SO_2$ in the greater area, peak during January (probably due
to increased heating in winter and the limited oxidation processes due to lower incoming solar
radiation) and are higher during spring months compared to summer. In general, the variation of
pollutant concentrations and the condensation sink is not great for the spring and summer seasons.
The only variable out of the ones considered that may explain the seasonality of NPF events at the
site is the increased concentrations of $PM_{10}$ found for spring months, which might be associated
with road sanding and salting that takes place in Scandinavian countries during the colder months
(Kupiainen et al., 2016) and are released in the ambient air during spring months (Stojiljkovic et al.,
2019).  The source of these particles though is uncertain, as no major differences in the wind roses
are found between the two seasons. Another study by Sarnela et al. (2015) at a different site in
southern Finland attributed the seasonality of NPF events in Finland to the absence of $H_2SO_4$
clusters during summer months due to a possible lack of stabilizing agents (e.g. ammonia). This



could explain the limited number of small particles (smaller than 10 nm) at the background sites
during summer.  In the more polluted environment at a roadside these agents may exist, but such
data was unfortunately not available.

Finally, a feature mentioned by Hao et al. (2018) in their study at the site of Hyytiälä, in which late
particle growth is observed was also found in this study. This happened on about 20% of NPF days
at FINRU (and a number of non-event days) and in most cases in early spring (before mid-April) or
late autumn (after mid-September). New particles were formed and either did not grow or grew very
slowly until later in the day when growth rates increased (Figure S2). In all these cases, growth
started when solar radiation was very low or zero, which probably associates the growth of particles
with nighttime chemistry leading to the formation of organonitrates (as found by the same study). A
similar behaviour was also rarely found at FINUB. Particle growth at late hours is not a unique
feature, as it was found at all sites studied. What is different in the specific events is the lack or very
slow growth during the daytime. Lower temperature (-0.81°C), incoming solar radiation (112 Wm$^{-2}$)
and higher relative humidity (68.4%) occurred on event days with later growth, while no clear wind
association was found. Lower concentrations of organic matter and nitrate were found throughout
the days with later growth compared to the rest of the NPF days. The very high average particle
number concentration in the smaller size bins is due to particles, though not growing to larger sizes
for some time, persisting in the local atmosphere for hours. These results though should be used
with caution due to the limited number of observations.





### 3.4 Spain

For Spain, data was available for an urban and a rural background site in the greater area of

Barcelona for the period 2012 - 2015. NPF events were rather frequent, occurring on about 12% of

the days at the rural site and 13.1% at the urban site. Though the sites are in close proximity (about

50 km), the seasonality of NPF events was different between them, peaking in spring at SPARU and

autumn at SPAUB (Figure 2). The frequency of NPF events in winter was relatively high compared

to the sites in central and northern Europe and higher than summer for both sites. Similarly, the

growth rate was similar for the two sites, being $3.62 \pm 1.86$ nm h$^{-1}$ at SPARU and $3.38 \pm 1.53$ nm h$^{-1}$ at SPAUB, again being higher in autumn for the urban site (which appears to be a feature of more

polluted sites), while the rural site follows the general trend of rural background sites, peaking in

summer (Figure 5). The formation rate $J_{10}$ at SPAUB is comparable to the other urban background

sites (apart from GERUB) and it peaked in spring, while once again the peak at SPARU was found

in summer, similar to the other rural sites of this study apart from the Greek (Figures 6 and 7). For

both sites a higher probability for events was found on weekends compared to weekdays, though

this trend is stronger at SPAUB (Figure 4). On the other hand, at the urban site both the growth and

formation rates were higher on weekdays compared to weekends (both $p < 0.001$). While the

increased growth rate during weekdays may be associated with the increased presence of

condensable species due to increased anthropogenic activities, the increased formation rate might be

affected by the increased emissions during these days.



In general, the atmospheric conditions favouring NPF events at both sites are similar to most other
sites, with lower relative humidity and higher solar radiation and wind speed ($p < 0.001$ for wind
speed at SPAUB) (Table S1). The wind profile between the two sites is different, with mainly
northwesterly and southeasterly winds for SPARU (which seems to be affected by the local
topography), while a more balanced profile is found at SPAUB. For both sites, though, increased
probability for NPF events is found for westerly and northwesterly winds. For both sites, these
incoming wind directions originate from a rather clean area with low concentrations of pollutants
and condensation sink (Table S4). At SPARU, incoming wind from directions with higher
concentrations of pollutants and condensation sink were associated with lower frequency of NPF
events but higher growth rates. At SPAUB, NPF events were relatively rare and growth rates were
lower with easterly wind directions, as air masses originating from that section have passed from
the city centre and the industrial areas from the Besos River. Due to this, incoming air masses from
these sectors had higher concentrations of pollutants and condensation sink. The concentrations of
all the pollutants with available data were lower at SPAUB (apart from $O_3$ and CO - the results for
the latter are not included) on NPF event days (Table S2) as was found by Brines et al. (2015), as
were the condensation sink and PM concentrations. At SPARU, the concentrations of the pollutants
with available data are rather low and as a result minimal differences were found between event and
non-event days.





While NPF events with subsequent growth of the particles were rare during summer, cases of bursts
of particles in the smallest size range available were found to occur frequently, especially in August
and July (the month with the fewest NPF events, despite the favourable meteorological conditions).
In such cases, a new mode of particles appears in the smallest size available, persisting for many
hours though without clear growth (brief or no growth is only observed), as reported by Dall'Osto
et al. (2012). Due to the lack of growth of the particles these burst events do not qualify as NPF
events using the criteria set in the present study. These burst events are associated with southerly
winds (known as Garbí-southwest and Migjorn-south in Catalan, which are common during the
summer in the area) that bring a large number of particles smaller than 30 nm to the site from the
nearby airport (located about 15 km to the southwest) and port (7 km south), as well as Saharan
dust, increasing the concentrations of PM (Rodríguez et al., 2001) and thus suppressing NPF events
due to the increased condensation sink.

Finally, the wind direction profile at SPARU appears to have a daily trend, with almost exclusively
stronger southeasterly winds at about midday (Figure S3), which might be the result of the
movement of the air masses due to the increased solar activity during that time (which results in
different heating patterns of the various land types in the greater area). These incoming southeast
winds are more polluted and have higher condensation sink, which almost consistently bring larger
particles at the site during the midday. This may explain to an extent the lowest probability for NPF
events from that sector, despite the very high concentrations of $O_3$ associated to them, with some





extreme values well above 100 µg m$^{-3}$ (Querol et al., 2017). The highest average growth rates are
also found from that direction.

**3.5    Greece**
Data are available for two background sites in Greece (2012 – 2018 for GRERU and 2015 – 2018
for GREUB), though not in close proximity. While in Greece meteorological conditions are
favourable in general for NPF events, with high solar radiation and low relative humidity, their
frequency was only about 8.5% for the urban background site in Athens and 6.5% for the rural
background site in Finokalia, similar to the frequency of Class I events in the study by Kalivitis et
al. (2019). Most NPF events occurred in spring at both sites, peaking in April (Figure 2). It is
interesting that all sites in southern Europe have a considerable number of NPF events during
winter, which might be due to the specific meteorological conditions found in this area, where
winter is a lot warmer than the sites in northern and central Europe. The growth rate of particles in
these events was found to be similar at both sites (3.68±1.41 nm h$^{-1}$ for GREUB and 3.78±2.01 nm
h$^{-1}$ for GRERU) and was higher in summer compared to the other seasons (Figures 3 and 5), having
a similar trend with the temperature and particulate organic carbon concentrations in the area. $J_{10}$
presented an interesting trend, having high averages in winter for both sites. Interestingly, the
lowest average $J_{10}$ was found for summer at both sites (Figure 7).





Similar to all sites, higher solar radiation and lower relative humidity compared to average
conditions were found on NPF event days (Table S1). Temperature and wind speed were found to
be lower, but the differences are minimal and are associated with the seasonal variability of the
events. The wind rose in GREUB mainly consists of northeasterly and southwesterly winds. Due to
its position, the site is heavily affected by emissions in Athens city centre with westerly winds,
resulting in increased particle number concentrations and condensation sink. Despite this, the
highest NPF probability and growth rates were found with northwesterly wind directions (Table
S4). This may be due to them being associated with the highest solar radiation (probably the result
of seasonal and diurnal variation), temperature and the lowest relative humidity, along with the
highest condensation sink and particle number concentrations of almost all sizes. Chemical
composition data was not available for GREUB, though $SO_2$ concentrations are rather low in
Athens and kept declining after the economic crisis (Vrekoussis et al., 2013). The seasonality of
$SO_2$ concentration in Athens favoured winter months and was at its lowest during summer for the
period studied (ΥΠΕΚΑ, 2012) (this trend changed later as $SO_2$ concentrations further declined),
which may also be a factor in the seasonality of NPF events, though this will be further discussed
later.

At the GRERU site, the wind profile is mainly westerly, and though it coincides with the most
important source of pollutants in the area, the city of Herakleio, its effect while observable is not
significant due to the topography in the area. The wind profile for NPF events is similar to the



average with significantly higher wind speeds (p < 0.001). In general, GRERU has very low
pollutant concentrations, with an average NO of 0.073 µg m⁻³, NO₂ of 0.52 µg m⁻³ and SO₂ in
concentrations below 1 ppb (Kouvarakis et al., 2002). Due to this, the differences in the chemical
composition in the atmosphere are also minimal (Table S2). For the specific site two different
patterns of development of NPF events were found. In one case, NPF events occurred in a rather
clear background, while in the other one they were accompanied with an increase in number
concentrations of larger particles or a new mode appearing at larger sizes (about a third of the
events). No differences were found in the seasonal variation between the two groups; increased
gaseous pollutant and particulate organic carbon concentrations were found for the second group
(though the differences were rather small) and a wind rose that favoured southwesterly winds
(originating from mainland Crete) instead of the northwesterly (originating from the sea) ones for
the first group. The growth rate for the two groups was found to be 3.56 nm h⁻¹ for the first group
and 4.17 nm h⁻¹ for the second, which might be due to the increased presence of condensable
compounds. As the dataset starts from the particle size of 8.77 nm, the possibility that these
particles were advected from nearby areas should not be overlooked, though they persisted and
grew at the site. Other than that, no significant differences were found for the different wind
directions.

As mentioned earlier, both sites had a very low frequency of events and $J_{10}$ in summer similar to
previous studies also reporting few or no events during summer (Vratolis et al., 2019; Ždímal et al.,




2011), though the incoming solar radiation is the highest and relative humidity is the lowest during
that season. This variation was also observed by Kalivitis et al. (2012) who associated the seasonal
variation of NPF events at GRERU to the concentrations of atmospheric ions. The effect of the
Etesian winds (known as Meltemia in Greek), which dominate the southern Aegean region during
the summer months though should not be overlooked. These result in very strong winds with an
average wind speed of 8.15 m s$^{-1}$ during summer at the Finokalia site, and increased turbulence
found in all years with available data, affecting both sites of this study. During this period, $N_{<30nm}$
drops to half or less compared to other seasons at both sites, while $N_{>100nm}$ is at its maximum due to
particle aging (Kalkavouras et al., 2017), increasing the condensation sink, especially in GRERU
(the effect in GREUB is less visible due to both the wind profile, blowing from east which is a less
polluted area, as well as the reduction of urban activities during summer months in Athens). Both
the increased condensation sink and turbulence are possible factors for the reduced number of NPF
events found at both sites in summer.  Another possible factor is the effect of high temperatures in
destabilising the molecular clusters critical to new particle formation.

**3.6    Region-Wide Events**
Region-wide events are NPF events which occur over large scale areas, that may cover hundreds of
kilometres (Shen et al., 2018). In the present study, NPF events that took place on the same day at
both background sites (urban background and rural) are considered as regional and their conditions
are studied (Table S5). The background sites in Greece were not considered due to the great





distance between them (about 350 km). There is also uncertainty for the background sites in
Finland, where the distance is about 190 km, though a large number of days were found when NPF
events occurred on the same day. The number of region-wide events per season (or the fraction of
region-wide events to total NPF events) is found in Figure 8 and it appears as if they are more
probable in spring at all the sites of the present study (apart from Finland, though the number of
events in winter was low), despite the differences found in absolute numbers.

In Denmark, about 20% of NPF events in DENRU were regional (the percentage is higher for
DENUB due to the smaller number of events, at 29%). The relatively low frequency of region-wide
NPF events can be explained by the different seasonal variation of NPF events (region-wide NPF
events were more frequent in spring compared to the average due to the seasonality of NPF events
in DENUB). Compared to local NPF event conditions, higher wind speed and solar radiation, as
well as $O_3$ and marine compound concentrations (results not included) were found, while the
concentrations of all pollutants (such as NO, $NO_x$, sulphate, elemental and organic carbon) were
lower. The exceptions found at DENRU (increased relative humidity and less incoming solar
radiation) are probably due to the different seasonality between local and region-wide NPF events at
the site, though region-wide events rarely present similar characteristics at different sites even in the
same country due to the differences in the initial meteorological and geographical conditions
(Hussein et al., 2009). The growth rates of region-wide events were found to be lower than those of
local events at both sites, which is probably associated with the limited concentrations of





condensable compounds due to the cleaner air masses of marine origin (as confirmed by the higher
concentrations of marine compounds).

In Germany, the majority of NPF events of this study were region-wide (about 60%). Compared to
the average, the meteorological conditions found for NPF event days compared to average
conditions were more distinct for the region-wide events, with even lower wind speed and relative
humidity and higher temperature and solar radiation, and all of these differences were significant (p
< 0.001). At GERRU where chemical composition data was available, higher concentrations of
particulate organic carbon and sulphate and lower nitrate concentrations were found. The
differences are significant (p < 0.001) and may explain the higher growth rates found in region-wide
events at both sites compared to the average, which is a unique feature. It should be noted that as
the majority of NPF events at the German sites are associated with easterly winds, it is expected that
in most cases the region-wide events will be associated with these, carrying the characteristics that
come along with them (increased growth rates and concentrations of organic carbon, as discussed in
Section 3.2).

In Finland, about a quarter of the NPF event days at FINRU (26%) occurred on the same day as at
FINUB (the frequency is a lot higher for FINUB, at 39%). As in Germany, the meteorological
conditions found on NPF event days compared to average conditions were more distinct during
region-wide events. Thus, for both sites temperature and relative humidity were lower while solar





radiation was higher. The different trend found for the wind speed at the two sites (being higher on
average NPF days at FINRU and lower at FINUB compared to average conditions) was enhanced
as well at the two sites for region-wide events. At FINRU where chemical composition data was
available, $NO_x$ and $SO_2$ had similar concentrations on region-wide event days, while $O_3$ was
significantly higher ($p < 0.001$). As at most other sites, the growth rate was found lower on region-
wide event days compared to the average at both sites.

Finally, in Spain the datasets of the two sites did not overlap greatly, having only 322 common
days. Among these days, 13 days presented with NPF events that took place simultaneously at both
sites, with smaller growth rates on average compared to local events (43% of the events at SPARU
and 36% of the events at SPAUB in the period 8/2012 to 1/2013 and 2014 when data for both sites
were available). Due to the small number of common events the results are quite mixed with the
only consistent result being the lower relative humidity and higher $O_3$ concentrations for regional
events at both sites, though none of these differences is significant. The wind profile at SPAUB
seems to further favour the cleaner sector, with the majority of incoming winds being from the NW
and even higher wind speeds (though with low significance). The result is similar at SPARU,
though less clear and with lower wind speeds.

These results are in general in agreement with those found in the UK in a previous study, where
meteorological conditions were more distinct on region-wide event days compared to local NPF





events; pollutant concentrations were lower as well as the growth rates of the newly formed
particles (Bousiotis et al., 2019).

Common events were also found between either of the background sites and the roadside, but they
were always fewer in number, due to the difference in their temporal variability compared to the
background sites, resulting from the effect of roadside pollution.

**3.7    The Effect of NPF Events on the Ultrafine Particle Concentrations**

The NSF is a metric of the effect of NPF events upon particle concentrations on either the days of
the events or over a larger timescale. Both the $NSF_{NUC}$ and $NSF_{GEN}$ were calculated for all sites of
this study and the results are presented in Figure 9. For almost all rural background sites $NSF_{NUC}$,
which indicates the effect of NPF on ultrafine particle concentrations on the day of the event, was
found to be greater than 2 (the only exception was GERRU), which means that NPF events more
than double the number of ultrafine particles (particles with diameter smaller than 100 nm)  at the
site on the days of the events, as NPF events are one of the main sources of ultrafine particles in this
type of sites, especially below 30 nm. This reaches up to 4.18 found at FINRU (418% more
ultrafine particles on the day of the events – 100% being the average), showing the great effect NPF
events have on rather clean areas. The long-term effect was smaller, and it was found that at FINRU
NPF events increase the number of ultrafine particles by about 130% in general. The effect of NPF
events was a lot smaller at the urban sites, though still significant at urban background sites





(reaching up 240% at FINUB on the days of events), while roadsides had the smallest NSF
compared to their respective background sites. This is because of the increased effect of local
sources such as traffic or heating, and the associated increased condensation sink found within these
sites, which cause the new particles to be scavenged by the more polluted background.

The calculation of NSF at the sites around Europe showed a weakness of the specific metric, which
points to the need for more careful interpretation of the results of this metric, especially at roadside
sites. At FINRO, the $NSF_{NUC}$ provided a value smaller than 1, which translates as ultrafine particles
are lost instead of formed on NPF event days. This though is the result of both the sharp reduction
in particle number concentrations at all modes that are required for NPF events to occur at a busy
roadside (much lower condensation sink), as well as a difference in the ratio between smaller to
larger particles (smaller or larger than 100 nm) on NPF event days (favouring the larger particles) at
the specific site.  Similarly, the long-term effect of NPF events at the site was found to be 1, which
means that NPF events appear to cause no changes in the number concentration of ultrafine
particles.

**4.    CONCLUSIONS**
There are different ways to assess occurrences of new particle formation (NPF) events. In this
study, the rate of NPF events, the growth rate of the particles and the frequency of NPF events,
associated with secondary formation of particles and not primary emissions, at 13 sites from five





countries in Europe are considered. The most consistent result found throughout the areas studied,
regardless of the geographical location was the higher frequency of NPF events at rural background
sites compared to roadsides. This pattern comes in contrast with what was found for the more
polluted Asian cities (Peng et al., 2017; Wang et al., 2017), where NPF events were more frequent
at the urban sites. This is probably associated with the even greater abundance of condensable
species associated with anthropogenic emissions, that promotes NPF events more, even compared
to the polluted cities in Europe. This contrast emphasises the differences in the occurrence of NPF
events between the polluted cities in Europe and Asia, which are associated with the level of
pollution found in them, as well as the influence that the level of pollution has on the occurrence of
NPF events. The type of site dependence found in Europe together with the average conditions
found on NPF event days compared to the average for each site, underline the importance of clear
atmospheric conditions at all types of site in Europe, especially for region-wide events (high solar
radiation and low relative humidity and pollutants concentrations). The temperature and wind speed
presented more diverse results which in many cases are associated with local conditions; the origin
of the incoming air masses though, appears to have a more important influence upon the NPF
events. Cleaner air masses tend to have higher probability for NPF events, while more polluted tend
to have higher growth rates (no consistent trend was found for the formation rate).

The frequency of NPF events at roadsides peaked in summer in all three countries with available
data. Greater variability in the seasonality of NPF events was found at the background sites. The





urban background sites presented more diverse results, for both the occurrence and development of
NPF events, especially compared to rural background sites. The within-week variation of NPF
events was found to favour weekends in most cases, as the pollution levels decrease, due to the
weekly cycle, especially at the roadsides. As background sites have smaller variations between
weekdays and weekends, the within-week variation of NPF events is smaller at the urban
background sites and almost non-existent at the rural background sites.

Both the growth rate of the newly formed particles and the formation rate of the particles were
found to be higher at all the roadsides compared to their respective rural and urban background.
While the more polluted urban environment is a limiting factor in the occurrence of NPF events,
their development as represented by the number of particles formed and the speed at which they
grow is enhanced by the urban environment (which seems to be a prerequisite for NPF events
within the more polluted environment), as more condensable compounds, deriving from
anthropogenic activities, are available. The picture is similar for $J_{10}$, the formation rate of particles
with 10 nm diameter (the rate of formed particles associated with NPF events that reached 10 nm
diameter), for which urban background sites were between their respective rural background sites
and the roadsides with the sole exception of DENUB (the difference with DENRU is rather small
though). The growth and formation rate at the rural background sites (apart from the Greek site)
was found to be higher in summer than in other seasons. On the other hand, the seasonality of the
growth rate at the roadsides is not clear but the formation rate peaks in the autumn at all three





roadside sites. While the trend at the rural sites is probably associated with the enhanced
photochemistry and increased concentrations of BVOCs during summer, the seasonality of the
growth rate at the roadside sites is more difficult to explain and probably shows the smaller
importance of the BVOCs compared to the compounds of anthropogenic origin (which are in less
abundance in summer) in this type of environment. In general though, higher temperatures were
associated with higher growth rates. This though applies only for the specific conditions at each site
and cannot be used as a general rule for the expected growth rate at a site, as locations with higher
temperatures did not present the higher growth rates.

While both the formation and growth rates are greater at the roadsides, the relative effect of NPF
events on the ultrafine particle concentrations is consistently a lot greater at the rural sites, where in
most cases NPF more than doubles (up to 400%) their particle number concentration on the days
they occur, as well as in the urban background sites where a substantial increase (up to 240%) is
also observed. The effect is considerable at roadside sites as well, increasing the number of ultrafine
particles up to 126% on event days (which might be higher as the occurrence of NPF events at
roadsides is harder to detect), which is limited compared to background sites due to the stronger
effect of local sources influencing the particle number concentration.

NPF events are an important source of ultrafine particles in the atmosphere for all types of
environments and are an important factor in the air quality of a given area. The present study

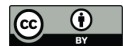



underlines the importance of both the synoptic and local conditions on NPF events, the mix of
which not only affects their development but can also influence their occurrence even in areas of
very close proximity. Since the mechanisms and general trends in NPF events are yet to be fully
explained and understood, more laboratory and field studies should be undertaken to generate new
knowledge.

**DATA ACCESSIBILITY**
Data supporting this publication are openly available from the UBIRA eData repository at
https://doi.org/10.25500/edata.bham.00000467

**AUTHOR CONTRIBUTIONS**
The study was conceived and planned by MDO and RMH who also contributed to the final
manuscript. The data analysis was carried out by DB who also prepared the first draft of the
manuscript. AM, JKN, CN, JVN, HP, NP, AA, GK, SV and KE have provided with the data for the
analysis. FDP, XQ, DCB and TP provided advice on the analysis.

**COMPETING INTERESTS**
The authors have no conflict of interests.




## 733 ACKNOWLEDGMENTS

This work was supported by the National Centre for Atmospheric Science funded by the U.K.
Natural Environment Research Council (R8/H12/83/011).





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





**TABLE LEGENDS:**

**Table 1:**   Location and data availability of the sites in the present study (RU denotes rural site, UB is urban background and RO is roadside).

**FIGURE LEGENDS**

**Figure 1:**   Map of the areas of study.

**Figure 2:**   Frequency (top panel) and seasonal (lower panel) variation of New Particle Formation events (Winter – DJF; Spring – MAM; Summer – JJA; Autumn – SON). For site naming first three letters refer to the country (DEN = Denmark, GER = Germany, FIN = Finland, SPA = Spain, GRE = Greece) while next two to the type of the site (RU = Rural Background, UB = Urban Background, RO = Roadside)

**Figure 3:**   Growth rate of particles up to 30 nm (with standard errors of the mean) during New Particle Formation events at all sites.

**Figure 4:**   Ratio of New Particle Formation event probability between weekends to weekdays. The greater the ratio the more probable it is for an event to take place during weekends compared to weekdays.

**Figure 5:**   Seasonal variation of growth rate of particles up to 30 nm during New Particle Formation events at all sites.

**Figure 6:**   Formation rate of 10 nm particles ($J_{10}$) (with standard errors of the mean) during New Particle Formation events at all sites.

**Figure 7:**   Seasonal variation of formation rate of 10 nm particles ($J_{10}$) during New Particle Formation events at all sites.

**Figure 8:**   Number of region-wide New Particle Formation events per season (top panel) and fraction of region-wide events to total New Particle Formation events per season for each site. Region-wide events are defined as those that occur on the same day at both background sites (Rural and Urban background).

**Figure 9:**   $NSF_{NUC}$ (average relative increase of ultrafine particles – particles of diameter up to 100 nm) due to New Particle Formation events on event days) and $NSF_{GEN}$ (average annual relative increase of ultrafine particles due to New Particle Formation events) at all sites.



**Table 1:** Location and data availability of the sites in the present study (RU denotes rural site, UB is urban background and RO is roadside).

| Site | Location | Available data | Meteorological data location | Data availability | Reference |
|------|----------|----------------|------------------------------|-------------------|-----------|
| DENRU | Lille Valby, 25 km W of Copenhagen, (55° 41' 41" N; 12° 7' 7" E) (2008 – 6/2010) Risø, 7 km north of Lille Valby, (55° 38' 40" N; 12° 5' 19" E) (7/2010 – 2017) | DMPS and CPC (5.8 - 700 nm, 89.3% availability), NO, $NO_x$, $SO_2$, $O_3$, minerals, OC, EC, $NO_3^-$, $SO_4^{2-}$, $NH_4^+$ | Ørsted – Institute station | 2008 – 2017 | Ketzel et al., 2004 |
| DENUB | Ørsted - Institude, 2 km NE of the city centre, Copenhagen, Denmark (55° 42' 1" N; 12° 33' 41" E) | DMPS and CPC (5.8 - 700 nm, 61.4% availability), NO, $NO_x$, $O_3$, minerals, EC | On site | 2008 – 2017 | Wang et al., 2010 |
| DENRO | H.C. Andersens Boulevard, Copenhagen, Denmark (55° 40' 28" N; 12° 34' 16" E) | DMPS and CPC (5.8 - 700 nm, 65.7% availability), NO, $NO_x$, $SO_2$, $O_3$, minerals, OC, EC, $NO_3^-$, $SO_4^{2-}$, $NH_4^+$ | Ørsted – Institute station | 2008 – 2017 | Wang et al., 2010 |
| GERRU | Melpitz, 40 km NE of Leipzig, Germany (51° 31' 31.85" N; 12° 26' 40.30" E) | TDMPS with CPC (4.8 - 800 nm, 90.4% availability), OC, $NO_3^-$, $SO_4^{2-}$, $NH_4^+$, $Cl^-$ | On site | 2008 – 2011 | Birmili et al., 2016 |
| GERUB | Tropos, 3 km NE from the city centre of Leipzig, Germany (51° 21' 9.1" N; 12° 26' 5.1" E) | TDMPS with CPC (3 - 800 nm, 68.3% availability) | On site | 2008 – 2011 | Birmili et al., 2016 |
| GERRO | Eisenbahnstraße, Leipzig, Germany (51° 20' 43.80" N; 12° 24' 28.35" E) | TDMPS with CPC (4 - 800 nm, 65.1% availability) | Tropos station | 2008 – 2011 | Birmili et al., 2016 |
| FINRU | Hyytiälä, 250 km N of Helsinki, Finland (61° 50' 50.70" N; 24° 17' 41.20" E) | TDMPS with CPC (3 – 1000 nm, 98.2% availability), NO, $NO_x$, $SO_2$, $O_3$, CO, $CH_4$, VOCs, $H_2SO_4$ | On site | 2008 – 2011 & 2015 – 2018 | Aalto et al., 2001 |
| FINUB | Kumpula Campus 4 km N of the city centre, Helsinki, Finland (60° 12' 10.52" N; 24° 57' 40.20" E) | TDMPS with CPC (3.4 - 1000 nm, 99.7% availability) | On site | 2008 – 2011 & 2015 – 2018 | Järvi et al., 2009 |
| FINRO | Mäkelänkatu street, Helsinki, Finland (60° 11' 47.57" N; 24° 57' 6.01" E) | DMPS (6 - 800 nm, 90.0% availability), NO, $NO_2$, $NO_x$, $O_3$, BC and $SO_2$ from Kalio Station | Pasila station and on site | 2015 – 2018 | Hietikko et al., 2018 |
| SPARU | Montseny, 50 km NNE from Barcelona, Spain (41° 46' 45" N; 2° 21' 29" E) | SMPS (9 – 856 nm, 53.7% availability), NO, $NO_2$, $SO_2$, $O_3$, CO, OM, $SO_4^{2-}$ | On site | 2012 - 2015 | Dall'Osto et al., 2013 |
| SPAUB | Palau Reial, Barcelona, Spain (41° 23' 14" N; 2° 6' 56" E) | SMPS (10.9 – 478 nm, 67.9% availability), NO, $NO_2$, $SO_2$, $O_3$, CO, BC, OM, $SO_4^{2-}$, $PM_{2.5}$, $PM_{10}$ | On site | 2012 – 2015 | Dall'Osto et al., 2012 |



| GRERU | Finokalia, 70 km E of Heraklion, Greece (35º 20' 16.8" N; 25º 40' 8.4" E) | SMPS (8.77 - 849 nm, 85.0% availability), NO, NO$_2$, O$_3$, OC, EC | On site | 2012 – 2018 | Kalkavouras et al., 2017 |
| GREUB | "Demokritos", 12 km NE from the city centre, Athens, Greece (37º 59' 41.96" N; 23º 48' 57.56" E) | SMPS (10 – 550 nm, 88.0% availability) | On site | 2015 – 2018 | Vassilakos et al., 2005 |





**Figure 1:** Map of the areas of study.





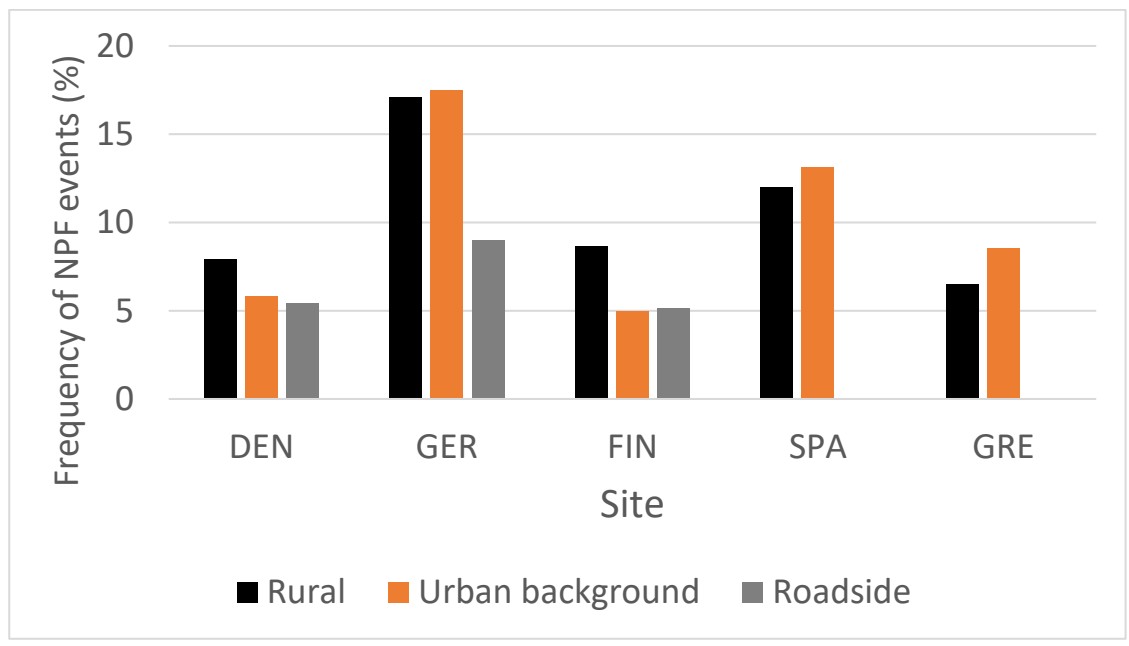

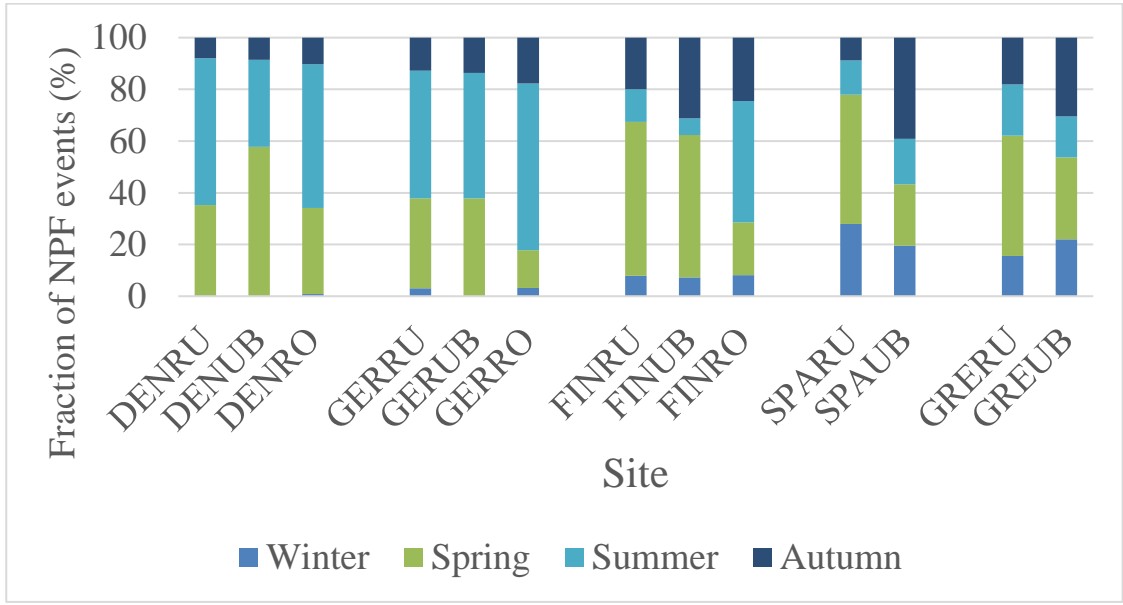

**Figure 2:** Frequency (top panel) and seasonal (lower panel) variation of New Particle Formation events (Winter – DJF; Spring – MAM; Summer – JJA; Autumn – SON). For site naming first three letters refer to the country (DEN = Denmark, GER = Germany, FIN = Finland, SPA = Spain, GRE = Greece) while next two to the type of the site (RU = Rural background, UB = Urban background, RO = Roadside)





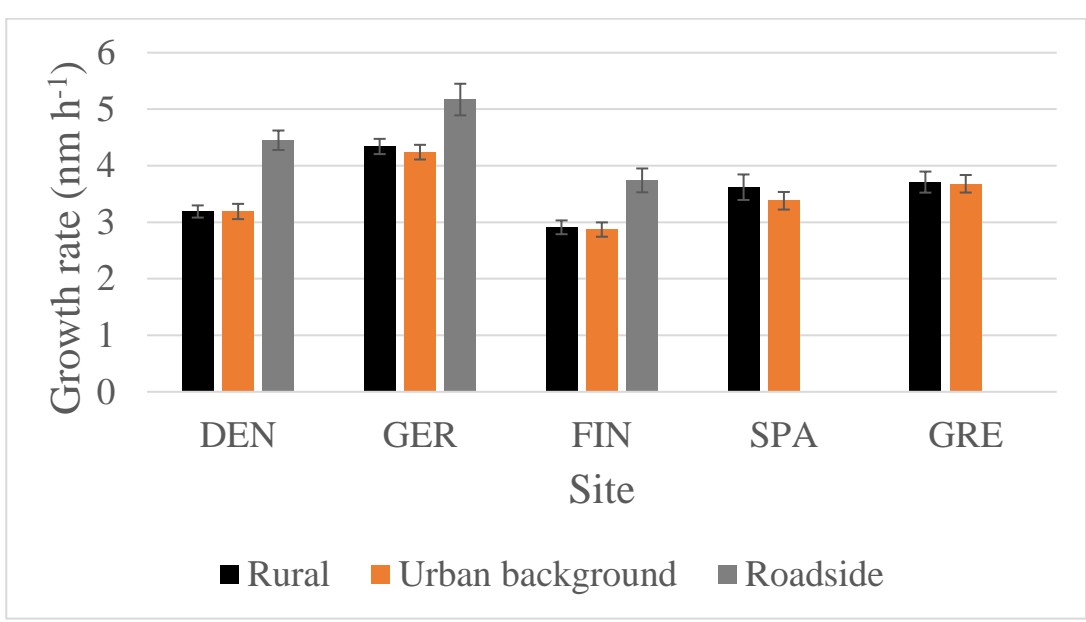

**Figure 3:** Growth rate of particles up to 30 nm (with standard errors of the mean) during New Particle Formation events at all sites.





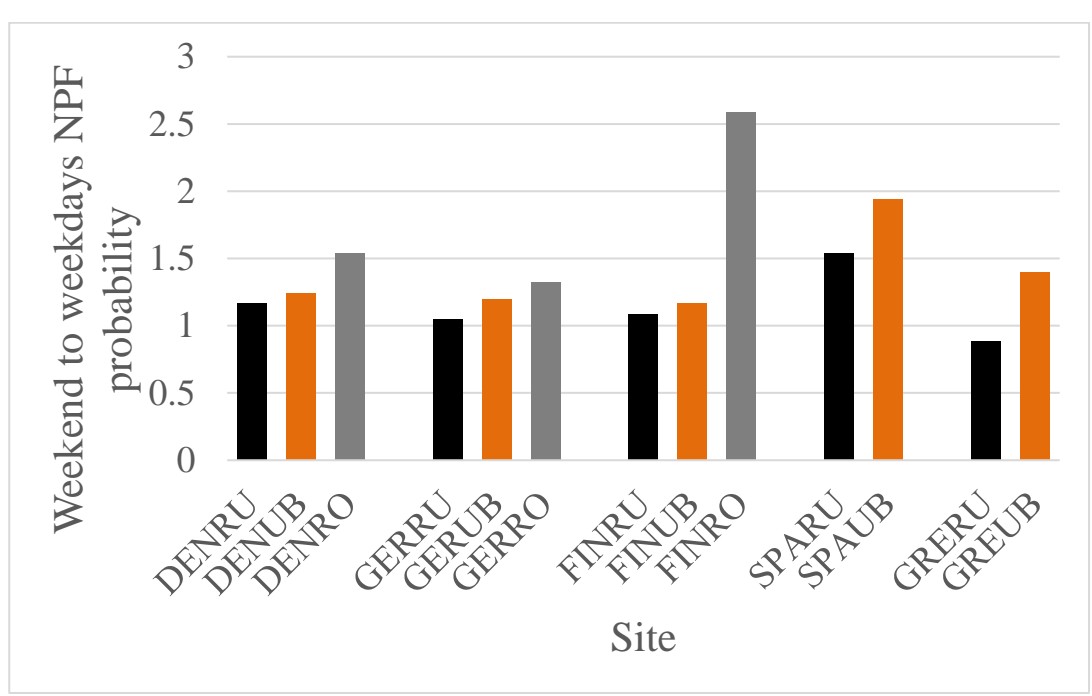

**Figure 4:** Ratio of New Particle Formation event probability between weekends to weekdays. The greater the ratio the more probable it is for an event to take place during weekends compared to weekdays.



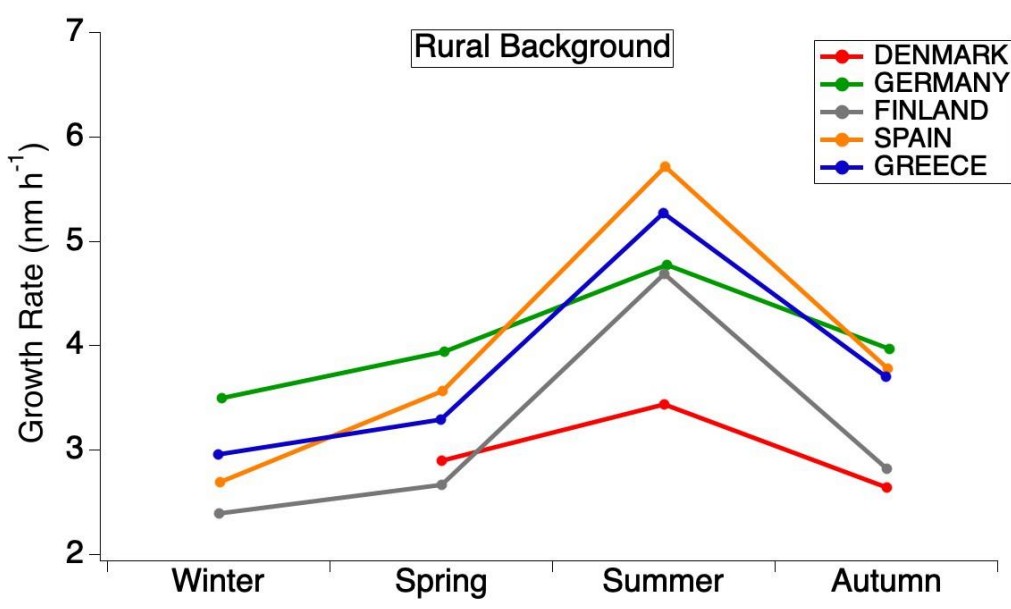

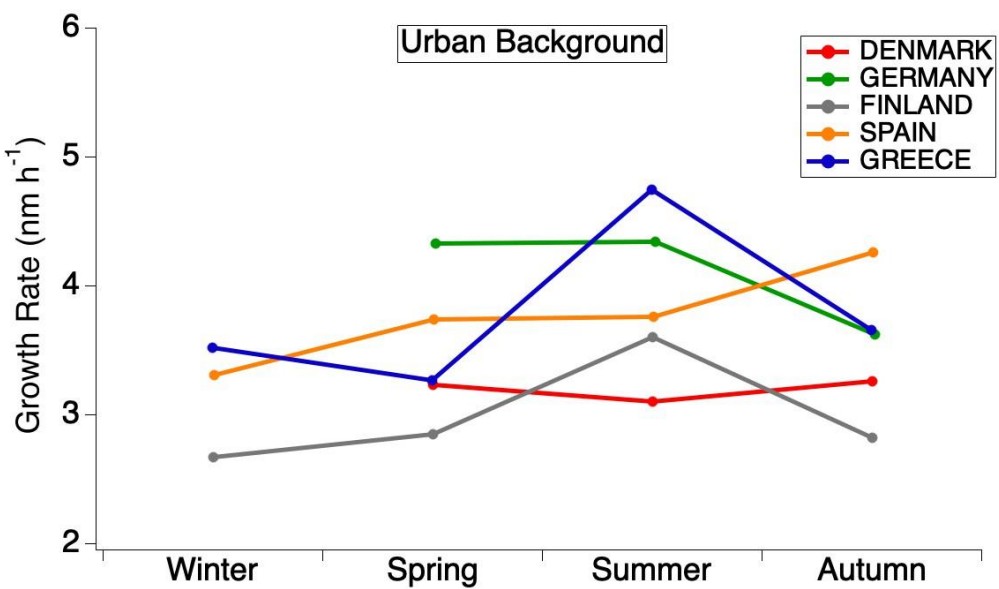





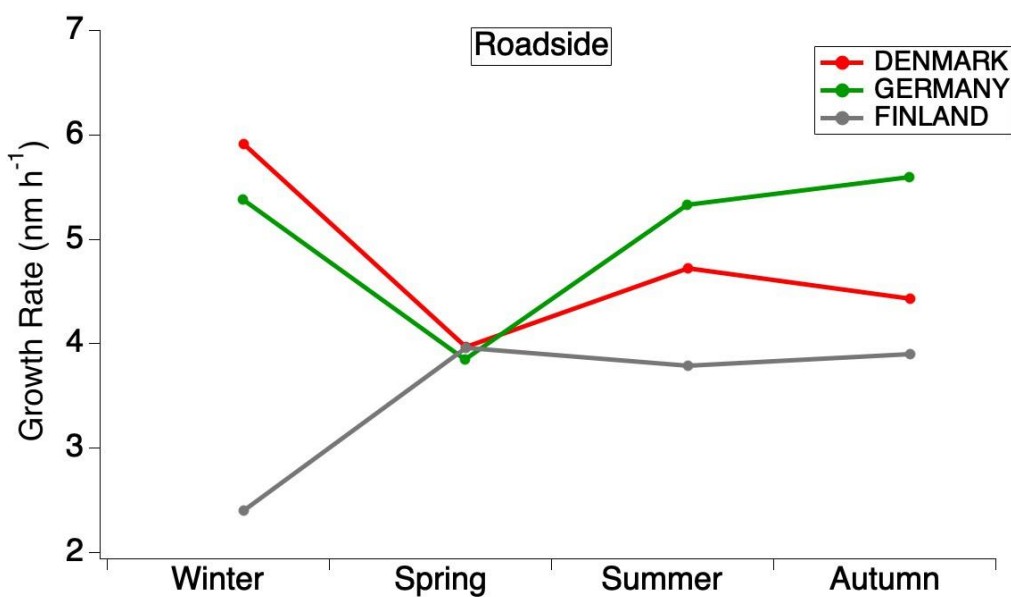

**Figure 5:** Seasonal variation of growth rate of particles up to 30 nm during New Particle Formation events at all sites.





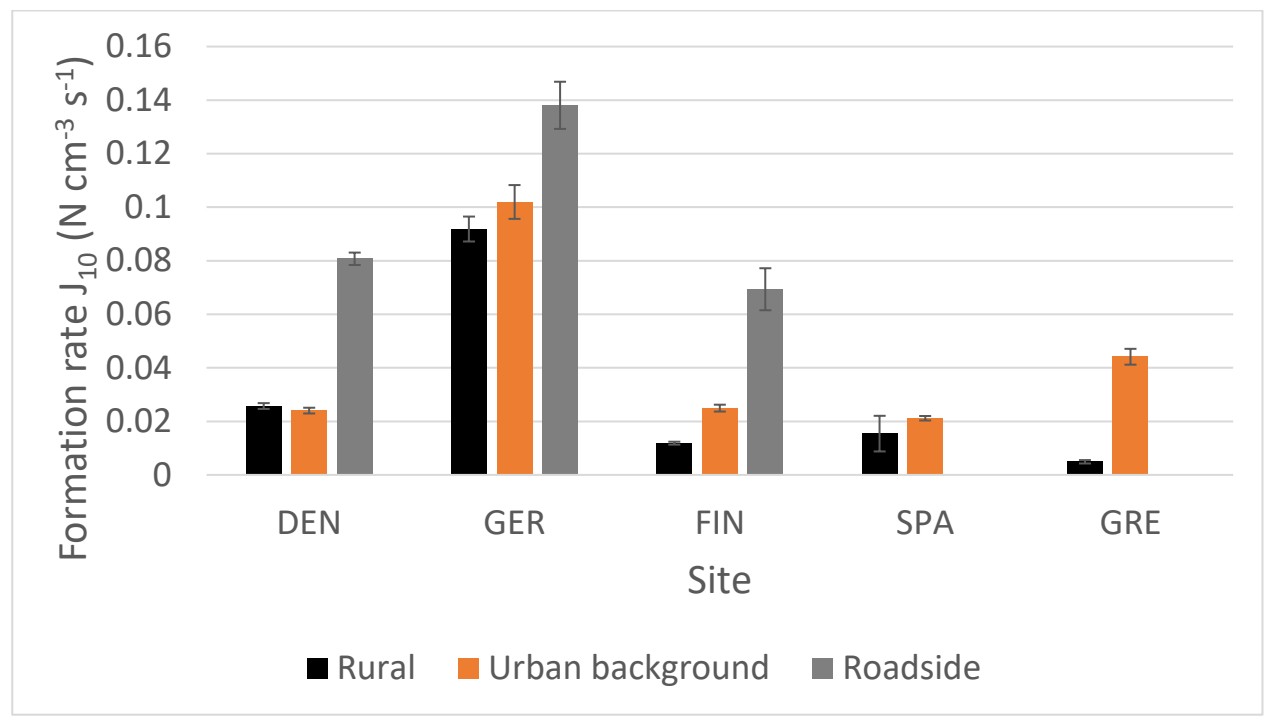

**Figure 6:** Formation rate of 10 nm particles ($J_{10}$) (with standard errors of the mean) during New
Particle Formation events at all sites.





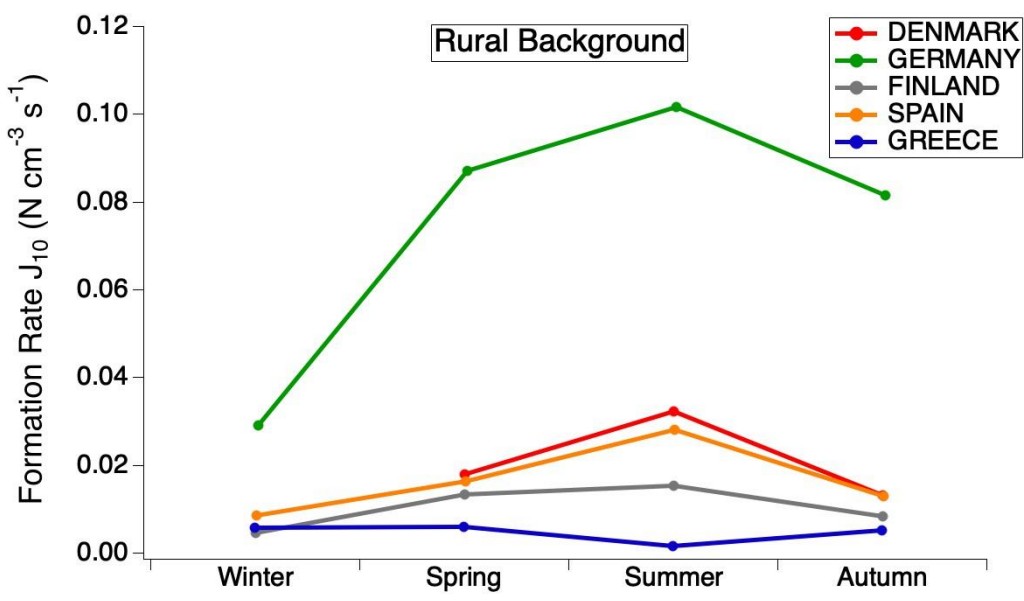

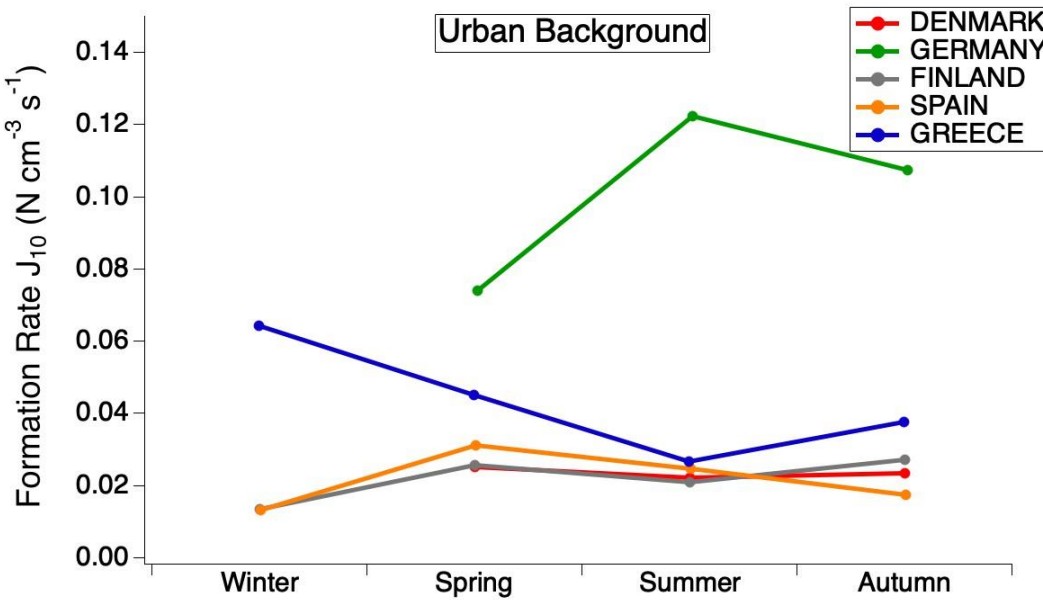





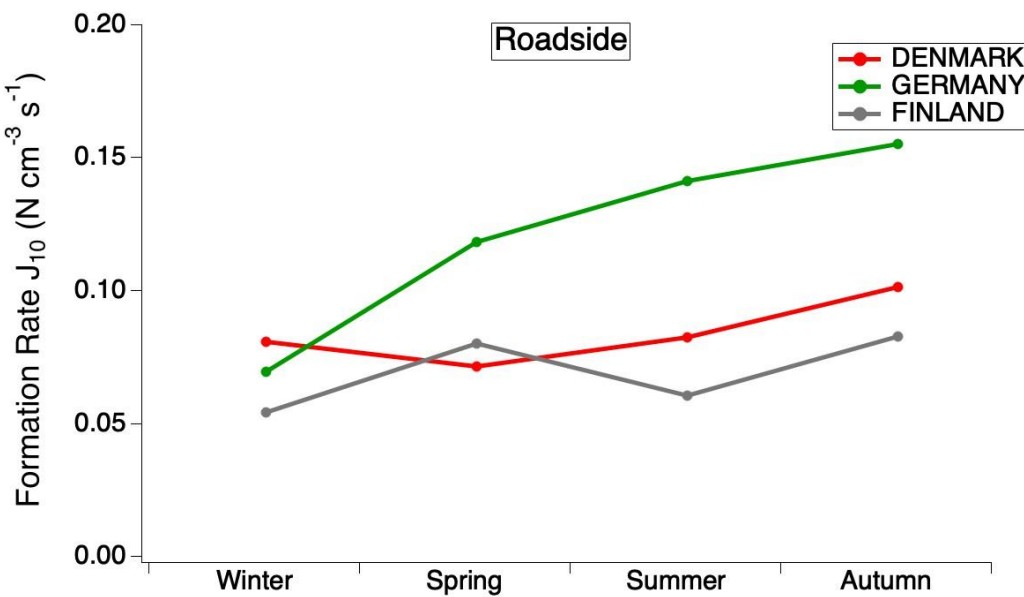

**Figure 7:** Seasonal variation of formation rate of 10 nm particles ($J_{10}$) during New Particle Formation events at all sites.





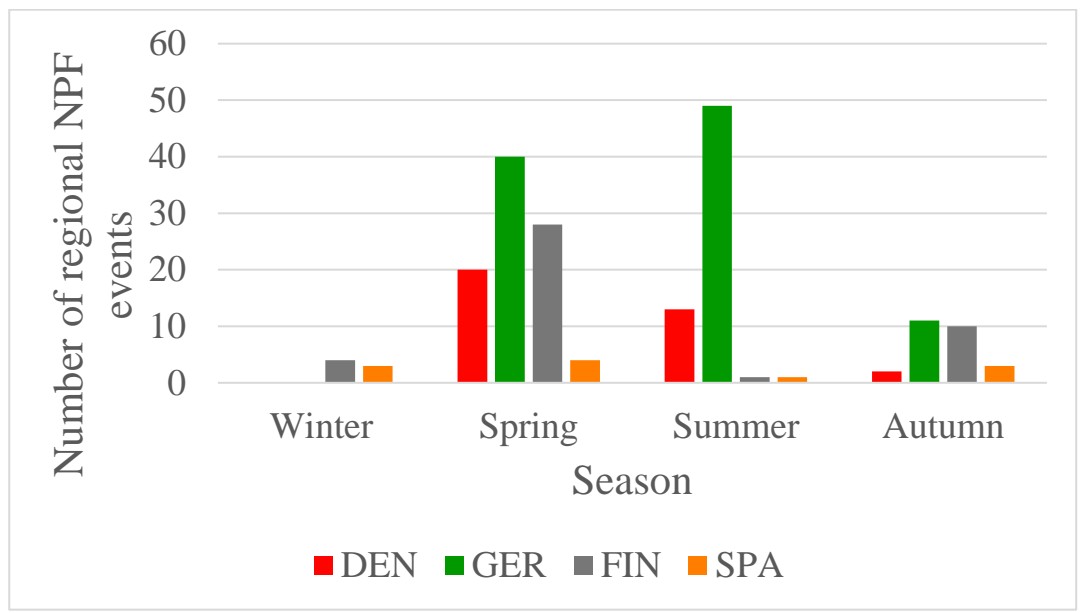

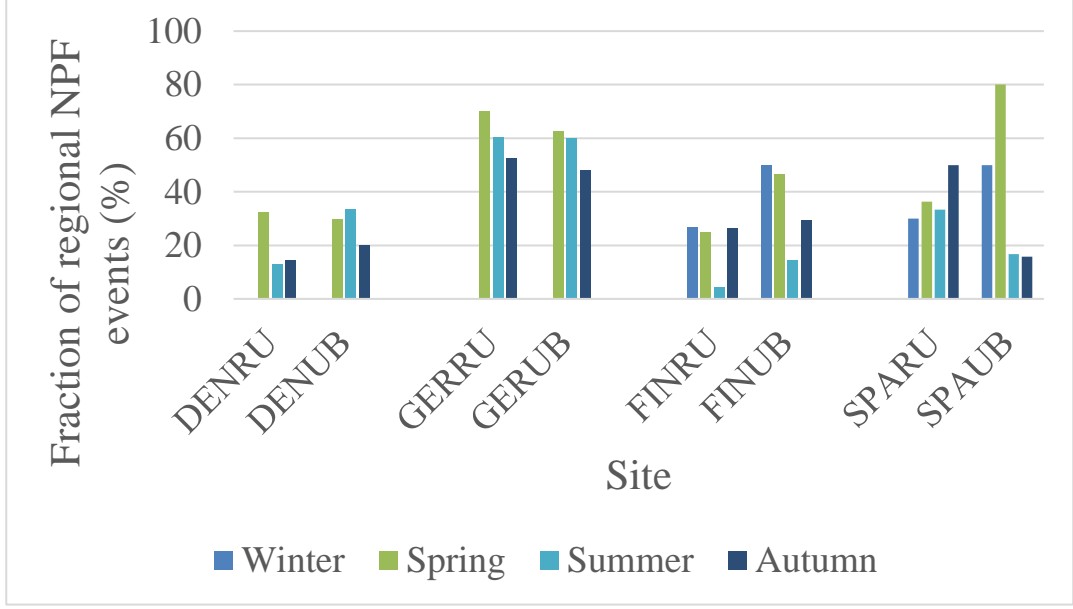

**Figure 8:** Number of region-wide New Particle Formation events per season (top panel) and fraction of region-wide events to total New Particle Formation events per season for each site. Region-wide events are defined as those that occur on the same day at both background sites (Rural and Urban background).





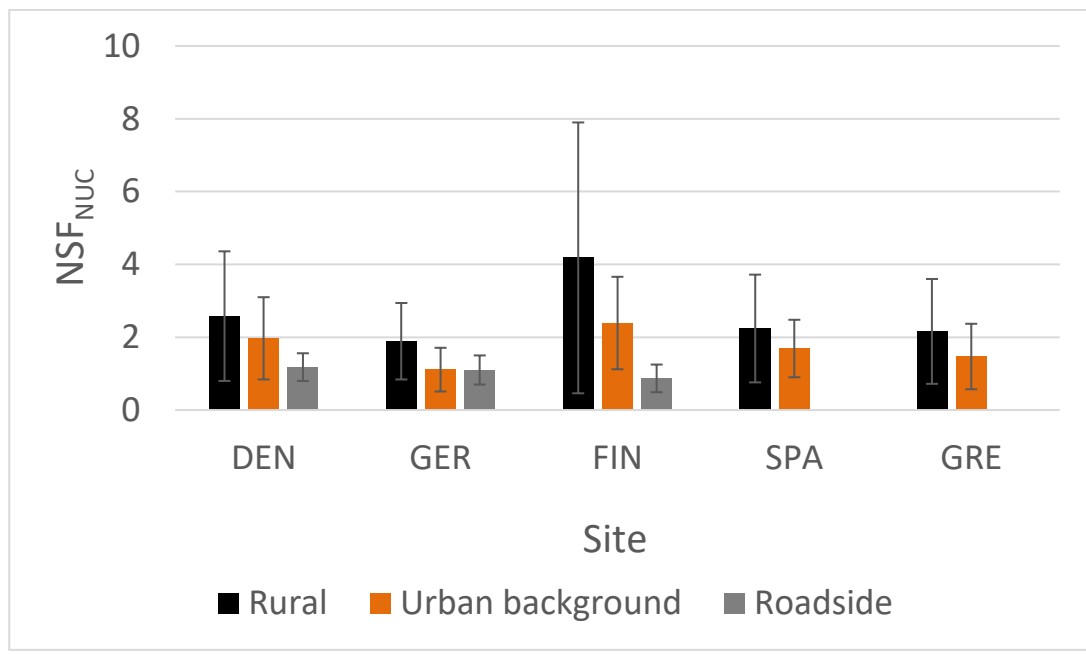

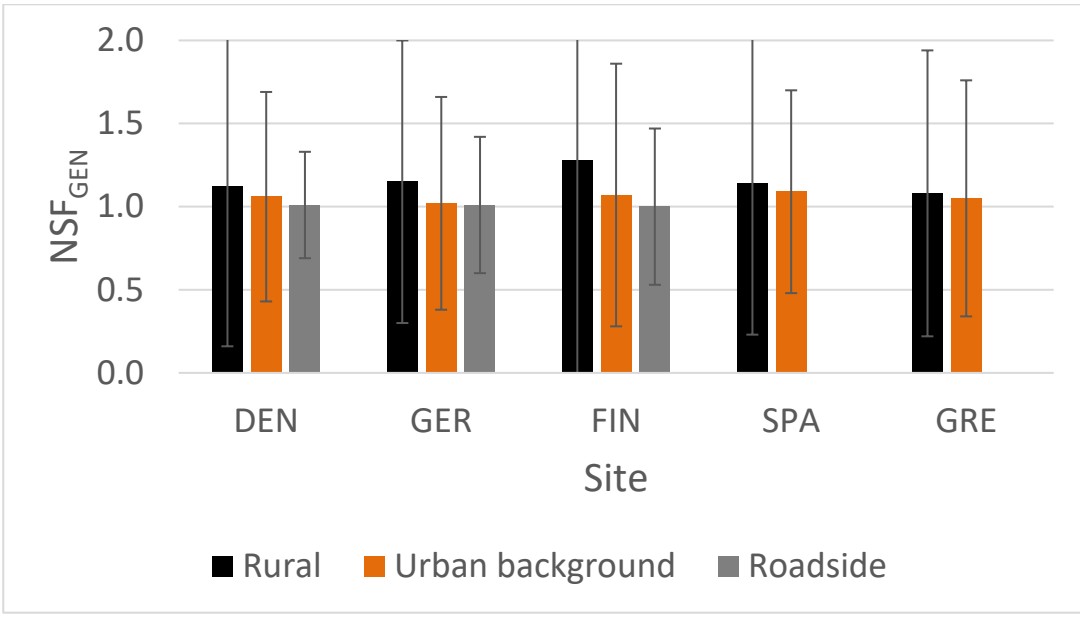

**Figure 9:** $NSF_{NUC}$ (average relative increase of ultrafine particles – particles of diameter up to 100
1370    nm) due to New Particle Formation events on event days) and $NSF_{GEN}$ (average annual relative
increase of ultrafine particles due to New Particle Formation events) at all sites.