# Peer review of "An Analysis of New Particle Formation (NPF) at Thirteen European Sites"

_Atmospheric Chemistry and Physics, 2020_

## Referee Comment (RC1) · Anonymous Referee #1 · 8 Oct 2020

This manuscript summaries long term measurements of particle number size distributions at thirteen European sites and derives the characteristics of new particle formation (NPF) events including NPF frequencies, formation rates of 10 nm particles, and growth rates of nanoparticles. While it is absolutely necessary to list the parameters for NPF events across Europe, it would be good to compare the numbers internally among the thirteen sites and externally with other continents, although the latter might not be necessary. Overall, I found that this manuscript a little bit hard to follow, since it is more like, or focus too much on, a combination of summaries of NPF events in a number of countries, which lacks the inherent connection between each part of the manuscript in the current format. At the moment, I would suggest the authors to address my following concerns.

[Figure]

1. This is actually my main concern and has been brought up in the general comments. Although the characteristics can be inferred from the available figures and tables, I would suggest the authors consider adding a general session, in addition to their conclusion, to summarize the NPF events in the thirteen sites/five countries. In fact, I believe some of the contents in the current conclusion should be moved into the new session. In this session, a quick idea on NPF events in these thirteen European sites can be established. If the authors want, they can even compare these events with those in other locations. The current description by countries, unfortunately, offers many details on concentrations of atmospheric tracers and meteorological parameters, and even the characteristics of NPF events in a particular location. This is OK but too descriptive, like a compilation of 5 papers. I believe that the details should appear in a manuscript that is specific for a location, instead a review like this one. I mean, one is less impressed by these without a logic or a general rule.

2. Elaborate how to exclude particles deriving from primary emissions next to pollution sources, such as traffic, which is important since the NPF frequencies are the highest at roadside sites in summer (Line 671).

3. Table 1 summaries the data availability of the thirteen sites. I noticed that some of the references do not match the available data, for example, the DENRU site, and thus assume the references are just a reference for the site instead of the data that have been discusses in the manuscript? Is it right? If yes, please add a footnote. Also, the size range for particles are not identical from one site to another, in contrast to the statement in Line 133. As a result, the range for particle growth rate calculation could be different, since "the size range between the minimum available particle diameter up to 30 nm" (Line 182) was used, which might lead to a minor correction in the growth rate. Lastly, these measurements were performed across at least 10 years, is there a distinguishable trend for NPF frequency, formation rate, and growth rate there?

4. The chemical composition of PM2.5 or PM10, in my mind, would not tell us a piece of evidence that is conclusive. I would rather use "CS" only to illustrate how particles

work as a scavenger. I don't believe that sulfate in PM2.5 is a good tracer to tell the mechanism.

5. The regional events are now defined as NPF events over hundreds of kilometers, and thus basically the manuscript is comparing two sites within one country. In addition to the statistics in the manuscript, have the authors tried to identify NPF events in an even larger distance? Clearly, we have to take wind direction and speed into account to make sure that what we are looking at are still "regional"? I would imagine that meteorology plays a big role here.

Minor comments,

6. It would be great, if the authors can use different colors to denote the site type in Figure 1. For those that are very close, please use a zoomed-in window. This practice may help understand the regional events later (Line 551).

7. Consider a better color-coding for the seasons in Figures 1 and 8.

---

## Referee Comment (RC2) · Anonymous Referee #3 · 24 Feb 2021

As the title suggests, the manuscript "An Analysis of New Particle Formation (NPF) at Thirteen European Sites" summarizes data on new particle formation events from a variety of sites. The manuscript focuses on describing how events differ based on location (urban, rural, roadside) and season. There is also discussion of regional events as well as an analysis of the contribution of NPF to ultrafine particle concentration.

NPF is a complex and poorly understood process. Studies such as this one that attempt to synthesize a large amount of data are valuable. However, these types of studies are inherently difficult owing to the complexity and uncertainties of npf. This may be what drives the largely descriptive nature of this paper. For this type of topic, I am not against a descriptive synthesis of information as I think it has its place and use. However, I think that the manuscript requires several major revisions before it is

suitable for publication.

Major comments

1) This manuscript attempts to bring together many datasets, a difficult undertaking and one that requires careful presentation of the data in order to clearly communicate the results. After careful reading, the new knowledge and major take-home points of the manuscript were unclear to me. This may, at least in part, be the result of the structure of the manuscript. I found the presentation difficult to follow, in part because the text and the figures were not harmonized; the text was organized by country, each figure targeted a specific result and included the data for all the countries. As a result, one has to keep moving between figures. With the current format, it is extremely difficult to identify overarching themes since any such results are buried along with discussion about local matters (that are nevertheless important) such as wind direction.

In my opinion, the manuscript would be greatly improved through restructuring Sects. 3.1-3.5 (the ones that focus on countries) to instead focus on results such as npf frequency, growth rate, etc. and for each section to discuss all the countries. This type of organization is already in place for Sects. 3.6 (regional events) and 3.7 (effect on ultrafine particle number). Alternatively, the sections could be organized based on location (urban, rural, background).

I also think a discussion section that summarizes the findings in a succinct way and clearly lays out the new findings and take-home points is warranted. Summarizing these along lines of "x is reduced under conditions of clean flow at all locations" would greatly improve readability and would allow one to judge the importance of the results more easily. To not add length to the paper, some of the more local details could be moved to the SI. Furthermore, some aspects of the conclusions (for instance, discussion comparing the results to results from some Asian cities) would be more appropriate in a section such as this one.

2) Although the manuscript is descriptive, it does draw comparisons between regions.

However, there is little to no discussion about how data limitations influence those comparisons. For instance, the sites in Germany have, on average, higher formation rates and frequency of events compared to the sites in other countries. Does this result still hold if the results are compared only to the results from 2008-2011 at the other sites (for those that have measurements)? In other words, is there significant interannual variability or are there trends that could affect the comparison between the countries given that the data coverage is not the same for each region? I am not suggesting that an in-depth analysis of trends or variability is required. I think a brief discussion about how such factors potentially impact the conclusions is sufficient.

Similarly, I would like to see some indication (table in SI) of the absolute number of events for each site for each season. That information is useful for the reader to interpret the results. Additionally, although data coverage is shown in Table 1, the current format doesn't provide information on if data coverage is biased towards specific seasons (which would impact how one interprets the seasonal results). From line 292, it sounds as if missing data may predominantly be in a few seasons at least for certain sites.

Other

1) Why are growth rates in Figure 3 provided as standard error of the mean, while in the text they appear to be standard deviations? It is also unclear to me if standard error of the mean is appropriate given the variability in drivers of npf and growth. This question also pertains to other figures where standard error of the mean is used – e.g. Fig. 6. This choice should be explained in the manuscript.

2) Some indication of variability is warranted in Figures 5, 7, and 8.

3) The explanation for how traffic related nucleation was removed from the data set (i.e. lines 156-157) is insufficient, particularly given the results shown in Fig. 6 that the formation rate is much higher at roadsides.

Technical

1) The naming of the sites (3 letter country, 2 letter location abbreviation) needs to be introduced in the text at the start of Sect. 3 not just in the figure caption.

2) The reference list should be checked for accuracy. For instance, the ACP rather than the ACPD version of Ketzel et al (2004) should be cited.

3) Figures are cited out of order (Fig. 5 before Fig. 4).

---

## Author Comment (AC1) · 18 Mar 2021

MS No.: acp-2020-414 Revised Title: A Phenomenology of New Particle Formation (NPF) at Thirteen European Sites Author(s): Bousiotis et al.

REFEREE #1 This manuscript summaries long term measurements of particle number size distributions at thirteen European sites and derives the characteristics of new particle formation (NPF) events including NPF frequencies, formation rates of 10 nm particles, and growth rates of nanoparticles. While it is absolutely necessary to list the parameters for NPF events across Europe, it would be good to compare the numbers internally among the thirteen sites and externally with other continents, although the latter might not be necessary. Overall, I found that this manuscript a little bit hard

to follow, since it is more like, or focus too much on, a combination of summaries of NPF events in a number of countries, which lacks the inherent connection between each part of the manuscript in the current format. At the moment, I would suggest the authors to address my following concerns.

1. This is actually my main concern and has been brought up in the general comments. Although the characteristics can be inferred from the available figures and tables, I would suggest the authors consider adding a general session, in addition to their conclusion, to summarize the NPF events in the thirteen sites/five countries. In fact. I believe some of the contents in the current conclusion should be moved into the new session. In this session, a quick idea on NPF events in these thirteen European sites can be established. If the authors want, they can even compare these events with those in other locations. The current description by countries, unfortunately, offers many details on concentrations of atmospheric tracers and meteorological parameters, and even the characteristics of NPF events in a particular location. This is OK but too descriptive, like a compilation of 5 papers. I believe that the details should appear in a manuscript that is specific for a location, instead a review like this one. I mean, one is less impressed by these without a logic or a general rule. RESPONSE: The Conclusions section has been reworked and reduced to now present conclusions about the results in general, forming a clearer take-home message. Additionally, a Discussion section has been added. This is separated in three parts: âÅć A section discussing the variability of the frequency, the seasonality and a summary of the effect of the atmospheric variables. Additionally, this includes the comparison between European and Asian sites and the justification for the differences found. ć A section discussing the variability and trends of the formation and growth rate among the sites, along with the variables that affect it. aĂć A section discussing the local conditions and the role they were found to play among the sites studied.

2. Elaborate how to exclude particles deriving from primary emissions next to pollution sources, such as traffic, which is important since the NPF frequencies are the highest

**ACPD**
at roadside sites in summer (Line 671). RESPONSE: It is not possible to exclude the particles deriving from primary emissions with the data available for this study. In order for something like this to be achieved additional traffic data are needed, as by subtracting the average (non-event) conditions would lead to negative formation rate values. In order to clarify what is presented and the difficulty to exclude the effect of traffic the following text was added in the Methods section: "As mentioned in the methodology for NPF event selection (chapter 2.2.1) days with particle formation associated with traffic emissions were excluded. For those extracted as NPF event days though, mainly for the roadside sites, such formation still occurs. It is impossible with the data available for this study to remove the traffic related particle formation in the calculations included in this study by effectively separating it from secondary particle formation or calculate it. Using average conditions for comparison would lead to negative values in most cases since in order for an NPE event to occur other emissions are reduced. This results in an overestimation of the formation rates at roadside sites presented in this study which, as mentioned earlier, was reduced as possible by choosing a time window for which we would have the maximum effect of secondary particle formation and the minimum possible effect from traffic related particle formation." (line 212)

3. Table 1 summaries the data availability of the thirteen sites. I noticed that some of the references do not match the available data, for example, the DENRU site, and thus assume the references are just a reference for the site instead of the data that have been discusses in the manuscript? Is it right? If yes, please add a footnote. Also, the size range for particles are not identical from one site to another, in contrast to the statement in Line 133. As a result, the range for particle growth rate calculation could be different, since "the size range between the minimum available particle diameter up to 30 nm" (Line 182) was used, which might lead to a minor correction in the growth rate. Lastly, these measurements were performed across at least 10 years, is there a distinguishable trend for NPF frequency, formation rate, and growth rate there? RESPONSE: A note that the referenced studies include a more detailed description of the sites was added at the table's legend. A note for the differences in the growth
rate was added in the Methods section: "Due to the differences in the smallest particle size available between the sites, a discrepancy would exist for the growth rate values presented (sites with lower size cut should present lower values of growth rate, as the growth rate tends to increase with particle size at this range). As a result, a direct comparison of the growth rate values found among sites with significant differences at the smallest particle size available was avoided." A long-term analysis for the metrics of NPF events was attempted but no significant trends were found. Nevertheless, tables of the values of the frequency, formation and growth rate per year for all the sites were added in the SI.

4. The chemical composition of PM2.5 or PM10, in my mind, would not tell us a piece of evidence that is conclusive. I would rather use "CS" only to illustrate how particles work as a scavenger. I don't believe that sulfate in PM2.5 is a good tracer to tell the mechanism. RESPONSE: Sulphate is mentioned three times in the text. ć Line 416, as one of the pollutants with elevated concentrations, discussing the possibility that slightly polluted conditions may have a positive effect on the occurrence of the events. ć Line 594, as one of the pollutants with lower concentrations on region-wide NPF events in Denmark thus acting a limiting factor for such events. As this may imply that sulphate may be one of the limiting factors the following note was added: "These cleaner atmospheric conditions are also confirmed by the lower CS on region-wide events, which is probably one of the most important factors in the occurrence of these large-scale events. " ć Line 611, as one of the pollutants with increased concentrations which can be a possible explanation for the higher growth rates found on region-wide events in Germany. Thus, we consider that the referee's concern is valid only for the second case, where the appropriate clarification was made.

5. The regional events are now defined as NPF events over hundreds of kilometers, and thus basically the manuscript is comparing two sites within one country. In addition to the statistics in the manuscript, have the authors tried to identify NPF events in an even larger distance? Clearly, we have to take wind direction and speed into account
to make sure that what we are looking at are still "regional"? I would imagine that meteorology plays a big role here. RESPONSE: Such an analysis was attempted. Sets of countries that could have common event days were studied (ex. Denmark and Germany). This though led to very few common days with NPF events and thus was not looked any further (with such a small number of events wrong assumptions would be made). Similar is the case with the possibility of regional events for the Greek sites, which even though they are in a distance for which regional events are considered, only a handful of common NPF event days were found.

Minor comments, 6. It would be great, if the authors can use different colors to denote the site type in Figure 1. For those that are very close, please use a zoomed-in window. This practice may help understand the regional events later (Line 551). RESPONSE: A new map has been designed with zoomed-in windows for all the sites in close proximity

7. Consider a better color-coding for the seasons in Figures 1 and 8. RESPONSE: The figures were updated with a clearer set of colours.

REFEREE #2 As the title suggests, the manuscript "An Analysis of New Particle Formation (NPF) at Thirteen European Sites" summarizes data on new particle formation events from a variety of sites. The manuscript focuses on describing how events differ based on lo- cation (urban, rural, roadside) and season. There is also discussion of regional events as well as an analysis of the contribution of NPF to ultrafine particle concentration. NPF is a complex and poorly understood process. Studies such as this one that at- tempt to synthesize a large amount of data are valuable. However, these types of studies are inherently difficult owing to the complexity and uncertainties of npf. This may be what drives the largely descriptive nature of this paper. For this type of topic, I am not against a descriptive synthesis of information as I think it has its place and use. However, I think that the manuscript requires several major revisions before it is

Major comments 1) This manuscript attempts to bring together many datasets, a dif-

**ACPD**
ficult undertaking and one that requires careful presentation of the data in order to clearly communicate the results. After careful reading, the new knowledge and major take-home points of the manuscript were unclear to me. This may, at least in part, be the result of the structure of the manuscript. I found the presentation difficult to follow, in part because the text and the figures were not harmonized; the text was organized by country, each figure targeted a specific result and included the data for all the countries. As a result, one has to keep moving between figures. With the current format, it is extremely difficult to identify overarching themes since any such results are buried along with discussion about local matters (that are nevertheless important) such as wind direction. In my opinion, the manuscript would be greatly improved through restructuring Sects. 3.1-3.5 (the ones that focus on countries) to instead focus on results such as npf frequency, growth rate, etc. and for each section to discuss all the countries. This type of organization is already in place for Sects. 3.6 (regional events) and 3.7 (effect on ultrafine particle number). Alternatively, the sections could be organized based on location (urban, rural, background). I also think a discussion section that summarizes the findings in a succinct way and clearly lays out the new findings and take-home points is warranted. Summarizing these along lines of "x is reduced under conditions of clean flow at all locations" would greatly improve readability and would allow one to judge the importance of the results more easily. To not add length to the paper, some of the more local details could be moved to the SI. Furthermore, some aspects of the conclusions (for instance, discussion comparing the results to results from some Asian cities) would be more appropriate in a section such as this one. RESPONSE: The results chapter has been restructured. It now consists of 5 sections presenting the results found: aÅć The frequency and seasonality of the events aÅć The formation and growth rates aĂć Conditions affecting NPF events aĂć Region-wide events aĂć The effect of NPF events on the ultrafine particle concentrations This structure is now easier to follow as it separates the results into smaller but consistent groups. Similarly, the figures now are associated with specific sections providing a better flow. Additionally, a new Discussion section was added (as mentioned earlier). In this the results found

**ACPD**
are summarised and general trends are pointed and discussed. The Conclusions part is also reworked, greatly reducing its size and providing with the take-home message.

2) Although the manuscript is descriptive, it does draw comparisons between regions. However, there is little to no discussion about how data limitations influence those comparisons. For instance, the sites in Germany have, on average, higher formation rates and frequency of events compared to the sites in other countries. Does this result still hold if the results are compared only to the results from 2008-2011 at the other sites (for those that have measurements)? In other words, is there significant interannual variability or are there trends that could affect the comparison between the countries given that the data coverage is not the same for each region? I am not suggesting that an in-depth analysis of trends or variability is required. I think a brief discussion about how such factors potentially impact the conclusions is sufficient. Similarly, I would like to see some indication (table in SI) of the absolute number of events for each site for each season. That information is useful for the reader to interpret the results. Additionally, although data coverage is shown in Table 1, the current format doesn't provide information on if data coverage is biased towards specific seasons (which would impact how one interprets the seasonal results). From line 292, it sounds as if missing data may predominantly be in a few seasons at least for certain sites. RESPONSE: The interannual variability was considered for the sites studied (especially those with longer datasets, namely the Danish, Finnish and the Greek rural sites). No significant variation was found, and no specific trend was observed for any of the metrics presented. To point these notes have been added in the Discussion section. Additionally, a table with the absolute number of NPF events per year, as well as the annual variation of the formation and growth rate were added for reference in the SI.

The possible bias caused by the seasonality of the data availability was looked into and only a slight change was found and noted in the text (events are almost equally frequent during spring and summer, though still spring remains the season with the highest frequency). A table was added in the SI with the NPF probability of the events
per season to provide a consistent metric of the seasonal frequency of the events, as well as another table with the seasonal data availability.

OTHER 1) Why are growth rates in Figure 3 provided as standard error of the mean, while in the text they appear to be standard deviations? It is also unclear to me if standard error of the mean is appropriate given the variability in drivers of npf and growth. This question also pertains to other figures where standard error of the mean is used – e.g. Fig. 6. This choice should be explained in the manuscript. RESPONSE: In all figures the standard deviation is now presented.

2) Some indication of variability is warranted in Figures 5, 7, and 8. RESPONSE: Adapted figures 5 and 7 to include standard deviations. This is not possible for figure 8 as it presents values without variation.

3) The explanation for how traffic related nucleation was removed from the data set (i.e. lines 156-157) is insufficient, particularly given the results shown in Fig. 6 that the formation rate is much higher at roadsides. RESPONSE: The traffic related particle formation could not be removed as it is impossible to separate or calculate it. NPF event days at roadside sites are days with reduced (though not inexistent) traffic emissions. The amount of this reduction is impossible to calculate though as it is impossible to separate primary emissions from secondary formation. Furthermore, the average conditions cannot be used as this would result in negative formation rates in many cases (due to the reduced emissions required for NPF events to occur). In order to clarify this the following text was added after discussing the formation rate calculation method: "As mentioned in the methodology for NPF event selection (chapter 2.2.1) days with particle formation associated with traffic emissions were excluded. For those extracted as NPF event days though, mainly for the roadside sites, such formation still occurs. It is impossible with the data available for this study to remove the traffic related particle formation in the calculations included in this by effectively separating it from secondary particle formation or calculate it. Using average conditions for comparison would lead to negative values in most cases since in order for an NPF event to occur other emis**ACPD**
sions are reduced. This results in an overestimation of the formation rates at roadside sites presented in this study which, as mentioned earlier, was reduced as possible by choosing a time window for which we would have the maximum effect of secondary particle formation and the minimum possible effect from traffic related particle formation." (line 212)

TECHNICAL 1) The naming of the sites (3 letter country, 2 letter location abbreviation) needs to be introduced in the text at the start of Sect. 3 not just in the figure caption. RESPONSE: The naming scheme is introduced at the Site description section (chapter 2.1) where they are first mentioned (line 135).

2) The reference list should be checked for accuracy. For instance, the ACP rather than the ACPD version of Ketzel et al (2004) should be cited. RESPONSE: References with wrong or old information has been updated.

3) Figures are cited out of order (Fig. 5 before Fig. 4). RESPONSE: Figures are now in the correct order

**ACPD**

---

## Author Response (AR2)

MS No.: acp-2020-414
**Revised Title: A Phenomenology of New Particle Formation (NPF) at Thirteen European Sites**
**Author(s): Bousiotis et al.**

**REVIEWER #1**
I thank the authors for their extensive revisions to the manuscript. The improvement to the structure makes the manuscript much easier to read and the addition of the separate discussion section more clearly articulates the findings of the manuscript. I recommend publication after consideration of the following minor details. Line numbers refer to the track changes version of the manuscript.

Lines 215-225: Thank you for the addition of this description. I find the information it presents to be helpful, however, the presentation of the information could be improved. For instance, in the second sentence "such formation still occurs" I believe is about traffic related npf, but the wording is unclear. I am also unsure what is meant by "using average conditions for comparison would lead to negative values" negative values of what? It isn't clear to me why the comparison would have a definitive sign. Finally, the last sentence would benefit from being broken into 2 sentences to increase readability.

**RESPONSE:** Text was updated for all cases suggested.
- For first one "associated with traffic emissions" was added.
- For the second an explanation for the reason negative values are expected was added. The text now reads "Using average conditions for comparison would lead to negative formation rate values in most cases, since in order for an NPF event to occur, traffic related particles are usually reduced to a greater extent compared to the formation from NPF, leading to lower particle concentrations on event days as found from a previous study in Marylebone Road, London (Bousiotis et al., 2019)."
- The final sentence was broken into two. The text now reads "This may result in an overestimation of the formation rates at roadside sites presented in this study. The choice of a time window for which we would have the maximum effect of secondary particle formation and the minimum possible effect from traffic related particle formation attempts to reduce this discrepancy as much as possible."

Table S6: I typically think of ND as standing for "not detected." Since here no measurements were made, it isn't exactly the same thing. I suggest a different abbreviation. The abbreviation should be defined in the caption.

**RESPONSE:** All ND abbreviations were replaced with NA as data were not available for these periods.

Line 1123-1124: "The most consistent result found throughout the areas studied, regardless of the geographical location, was the higher frequency of NPF events at rural background sites compared to roadsides." I wonder how consistent this result truly is. I suggest reconsidering the phrase "most consistent result…regardless of geographical region" for comparing rural background and roadsides. From Figure 2, only 3 of the countries had measurements at roadsides. For Finland, the results for urban background and roadside are quite similar. So that leaves 2 out of the 5 countries.

**RESPONSE:** The text was updated to clarify that the result is consistent among the countries of this study with available data for both types of sites. Text now reads "A higher frequency of NPF events at the rural background sites compared to roadsides was found for all countries with available data for both types of site." As for the Finnish sites, while the reviewer is correct about the similarity of the urban background site and the roadside, the text discusses the relation between the rural background sites and roadsides.

Lines 1151-1153 "Finally, it should be noted that no clear interannual trend…" I think this statement should be made significantly earlier in the manuscript so that the reader isn't wondering throughout the long

manuscript what role trends might play. I think a slightly revised version of this sentence would be appropriate at the beginning of section 3.

**RESPONSE:** To mention the lack of interannual variations, the sentence in the introductory part of section 3 was changed to: "The annual number of NPF events, growth rate and formation rate for all the sites is found in Table S6, for which no clear interannual trend is found for any of the sites in this study. This may be due to the relatively short period of time studied for such variations to be observed." (line 250).

**REVIEWER #2**

This manuscript provides a useful overview of new particle formation across Europe. To me the big-picture summary expressed in the figures is that, all things considered, formation and especially growth rates are remarkably uniform across time and space. Yes there is variability, but it is not so large. This is most notable for the growth rates. If the authors agree with this assertion, in my opinion this would be worth including in the abstract. Further, I suggest keeping the y axis range as constant as possible in the summary figures - most of these have similar maximum values (i.e. growth rates between 6 and 10 nm/h) and the presentation would profit from the immediate visual similarity in the ranges.

**RESPONSE:** The phrase "(though in many cases differences between the sites were small)" was added in the abstract (line 47). Additionally, Y-axis ranges in both growth and formation rate figures are now uniform across all figures.